# Improved Algorithms for Contextual Dynamic Pricing

**Matilde Tullii**[*]
FairPlay Team, CREST, ENSAE

**Solenne Gaucher**[*]
FairPlay Team, CREST, ENSAE

**Nadav Merlis**
FairPlay Team, CREST, ENSAE

**Vianney Perchet**
FairPlay Team, CREST, ENSAE - Criteo AI Lab

## Abstract

In contextual dynamic pricing, a seller sequentially prices goods based on contextual information. Buyers will purchase products only if the prices are below their valuations. The goal of the seller is to design a pricing strategy that collects as much revenue as possible. We focus on two different valuation models. The first assumes that valuations linearly depend on the context and are further distorted by noise. Under minor regularity assumptions, our algorithm achieves an optimal regret bound of $\tilde{\mathcal{O}}(T^{2/3})$, improving the existing results. The second model removes the linearity assumption, requiring only that the expected buyer valuation is $\beta$-Hölder in the context. For this model, our algorithm obtains a regret $\tilde{\mathcal{O}}(T^{d+2\beta/d+3\beta})$, where $d$ is the dimension of the context space.

## 1 Introduction

Setting a price and devising a strategy to dynamically adjust it poses a fundamental challenge in revenue management. This problem, known as dynamic pricing or online posted price auction, finds applications across various industries and has received significant attention from economists, operations researchers, statisticians, and machine learning communities. In this problem, a seller sequentially offers goods to arriving buyers by presenting a one-time offer at a specified price. If the offered price falls below the buyer's (unknown) valuation of the item, a transaction occurs, and the seller obtains the posted price as revenue. Conversely, if the price exceeds the buyer's valuation, the transaction fails, resulting in zero gain for the seller. Crucially, the seller solely receives binary feedback indicating whether the trade happened. Her objective is to learn from this limited feedback how to set prices that maximize her cumulative gains while ensuring that transactions take place. In this paper, we study the problem of designing an adaptive pricing strategy, when the seller can rely on contextual information, describing the product itself, the marketing environment, or the buyer.

While this problem has been extensively studied, previous results either rely on strong assumptions on the structure of the problem, greatly limiting the applicability of such approaches, or achieve sub-optimal regret bounds. In this work, we aim to improve both aspects—achieving better regret bounds while making minimal assumptions about the problem. Specifically, we study two different models for the valuation of buyers as a function of the context: *1) linear valuations*, where the item valuation of buyers is an unknown noisy linear function of the context; and *2) non-parametric valuations*, where the valuation is given by an unknown Hölder-continuous function of the contextual information, perturbed by noise.

---

[*]Equal contribution.

38th Conference on Neural Information Processing Systems (NeurIPS 2024).

Table 1: Summary of existing regret bounds. $g$ is the expected valuation function, $F$ is the c.d.f. of the noise, and $\pi(x, p)$ is the reward for price $p$ and context $x$, defined in Section 2.1.

| Model | Noise Assumption | Regret |
|---|---|---|
| Linear | $F$ is known | $\tilde{\mathcal{O}}(T^{2/3})$ [11] |
| | $F$ is known or parametric, and log-concave | $\tilde{\mathcal{O}}(T^{1/2})$ [15] |
| | $F$ has $m$-th order derivatives | $\tilde{\mathcal{O}}(T^{2m+1/4m-1})$ [14] |
| | $F''$ is bounded | $\tilde{\mathcal{O}}(T^{2/3}) \vee \|\theta - \widehat{\theta}\|_1 T$ [22] |
| | $F$ is Lipschitz | $\tilde{\mathcal{O}}(T^{3/4})$ [14, 21], $\tilde{\mathcal{O}}(T^{2/3})$ **[this work]** $\Omega(T^{2/3})$ [31] |
| | Bounded noise | $\tilde{\mathcal{O}}(T^{3/4})$, [31] |
| Non-parametric | $\pi(x, \cdot)$ is quadratic around its maximum for all $x$, $F$ and $g$ are Lipschitz | $\tilde{\mathcal{O}}(T^{d+2/d+4})$ [10] $\Omega(T^{d+2/d+4})$ [10] |
| | $F$ is Lipschitz and $g$ is Hölder | $\tilde{\mathcal{O}}(T^{d+2\beta/d+3\beta})$ **[this work]** |

## 1.1 Related Work

Dynamic pricing has been extensively studied for half a century [19, 26], leading to rich research on both theoretical and empirical fronts. For comprehensive surveys on the topic, we refer the readers to [6, 12]. While earlier works assumed that the buyer's valuations are i.i.d. [18, 5, 16, 9], recent research has increasingly focused on feature-based (or contextual) pricing problems. In this scenario, product value and pricing strategy depend on covariates. Pioneering works considered valuations depending deterministically on the covariates. Linear valuations have been the most studied [3, 15, 11, 20], yet a few authors have also explored non-parametric valuations [24].

Recent works have extended these methods to random valuations, mainly assuming that valuations are given by a function of the covariate, distorted by an additive i.i.d. noise. As this poses more challenges, authors have mostly focused on the simplest case of linear valuation functions, under additional assumptions. Initial studies assumed knowledge of the noise distribution [11, 15, 30]. This assumption was later relaxed, albeit with additional regularity requirements on the cumulative distribution function (c.d.f.) of the noise and/or the reward function [14, 22], and then again by [31], that achieves a regret bound of $\tilde{\mathcal{O}}(T^{3/4})$ for linear valuations, while assuming only the boundedness of the noise. Closest to our work [14, 21] also focus on the case in which the only regularity required is the Lipschitzness of the CDF. Their approaches show some similarities with our work but still achieve suboptimal regret rates. A more detailed comparison between ours an their algorithms is presented later on in the paper. Other parametric models have been explored, with, for example, generalized linear regression models [28], though they also require strong assumptions, including quadratic behavior of the reward function around each optimal price. Few works have considered non-deterministic valuations with non-parametric valuation functions. Among those, [10] consider Lipschitz-continuous valuation functions of $d$-dimensional covariates. They achieve a regret of order $\tilde{\mathcal{O}}(T^{d+2/d+4})$, assuming again quadratic behaviour around optimal prices. We refer to Table 1 for a comprehensive comparison between different previous works, their assumptions and regret bounds.

To improve on previous results, we design algorithms that share information on the noise distribution across different contexts. This idea relates to methods used in *cross-learning*, a research direction stemming from online bandit problems with graph feedback [23, 2]. In this framework, introduced by [4] and further studied in [27], when choosing to take action $i$ in context $x_t$, the agent observes the reward $r_i(x_t)$ along with rewards $r_i(x'_t)$ associated with other contexts $x_{t'}$. Our algorithms leverage similar principles to learn information usable across different contexts. However, compared to the typical problems addressed by cross-learning methods (e.g., first-price auctions, sleeping bandits, multi-armed bandits with exogenous costs), the contextual dynamic problem is more complex due to the intricate dependence of the reward on the unknown valuation function.

## 1.2 Outline and Contributions

In this work, we tackle the problem of dynamic pricing with contextual information. We consider two models for the expected valuations of the buyer, assuming respectively that they are given by a

linear function, or by a non-parametric function. For both models, we present a general algorithmic scheme called VALUATION APPROXIMATION - PRICE ELIMINATION (VAPE), and provide bounds on its regret in both models:

- In the linear model, we obtain a regret of $\tilde{O}(T^{2/3})$, assuming only that the c.d.f. of the noise is Lipschitz. This concludes an extensive series of papers on the topic, as it establishes the minimax optimal regret rate and proves it is attainable under minimal assumptions.

- In the non-parametric model, we obtain a regret rate of $\tilde{O}(T^{d+2\beta/d+3\beta})$, assuming only the Lipschitz-continuity of the noise and the Hölder one of the valuation function. This result is the first of its kind under such minimal assumptions.

The rest of the paper is organized as follows. We begin by presenting the model and summarizing the notations used throughout the paper in Section 2.1. Section 2.2 outlines our assumptions and compares them with those in previous works. In Section 2.3, we discuss the main sources of difficulty of the problem and highlight the importance of information sharing in contextual dynamic pricing. In Section 3, we present our algorithmic scheme, VAPE, and provide an initial informal result bounding its regret. Then, in Section 4, we apply this algorithmic scheme to linear valuations and provide a bound on its regret. Finally, in Section 5, we extend this algorithm to non-parametric valuations.

## 2 Preliminaries

### 2.1 Model and Notations

The problem of dynamic pricing with contextual information is formalized as follows. At each step $t \leq T$, a context $x_t \in \mathbb{R}^d$, describing a sale session (product, customer, and context) is revealed. The customer assigns a hidden valuation $y_t$ to the product, and the seller proposes a price $p_t$, based on $x_t$ and on historical sales records. If $p_t \leq y_t$, the trade is successful, and the seller receives a reward $y_t$; otherwise the trade fails. The seller's only feedback is the binary outcome $o_t = \mathbb{1}\{p_t \leq y_t\}$. We assume that the seller's valuation is given by

$$y_t = g(x_t) + \xi_t, \tag{1}$$

where $g : \mathbb{R}^d \mapsto \mathbb{R}$ is the valuation function, and $\xi_t$ is a centered, bounded, i.i.d. noise term, independent of $x_t$ and of $(x_s, p_s, \xi_s)_{s<t}$. In the present paper, we consider successively linear and non-parametric valuation functions $g$ in Sections 4 and 5. The seller's objective is to maximize the sum of her cumulative earnings. We denote by $\pi(p, x_t)$ the expected reward of the seller if she posts a price $p$ for a product described by covariate $x_t$:

$$\pi(x_t, p) = \mathbb{E}[p\mathbb{1}\{p \leq y_t\}|p, x_t].$$

Adopting the terminology of the literature on multi-armed bandits, we measure the performance of our algorithm and the difficulty of the problem through the regret $R_T$, defined as

$$R_T = \sum_{t=1}^{T} \max_{p \in \mathbb{R}} \pi(x_t, p) - \sum_{t=1}^{T} \pi(x_t, p_t).$$

**Notations** Throughout this paper, we make use of the following notation. We denote by $\|\cdot\|$ the Euclidean norm. For all $A, B \in \mathbb{R}$, we denote by $[\![A, B]\!]$ the set $\{A, A+1, \ldots, B\}$. $R_T \lesssim B_T$ (resp. $R_T = \tilde{\mathcal{O}}(B_T)$) means that there exists a (possibly problem-dependent) constant $C$ such that $R_T \leq CB_T$ (resp. $R_T = \mathcal{O}(\log(T)^C B_T)$). Finally, $f$ and $F$ denote the p.d.f. and c.d.f. of the noise, respectively.

### 2.2 Assumptions

For both valuation models, we make the following assumptions on the context and noise distribution.

**Assumption 1.** *Contexts and expected valuations are bounded:* $\|x_t\|_2 \leq B_x$ *and* $|g(x_t)| \leq B_g$ *a.s.*

This assumption is classical in contextual dynamic pricing problems. We underline that contexts do not need to be random. In particular, they can be chosen by an adaptive adversary, aware of the

seller's strategy, and based on past realizations of $(x_s, p_s, \xi_s)_{s<t}$. Assumption 1 is milder than the i.i.d. context assumption appearing in [14, 28, 10].

Dynamic pricing strategies mostly assume that the buyer's valuations are bounded. To enforce this, we assume that the noise is bounded; moreover, we assume that its c.d.f. Lipschitz continuous.

**Assumption 2.** *The noise $\xi_t$ is bounded: $|\xi_t| \leq B_\xi$ a.s. Moreover, its c.d.f. $F$ is $L_\xi$-Lispchitz continuous: for all $(\delta, \delta') \in \mathbb{R}^d$, $|F(\delta) - F(\delta')| \leq L_\xi |\delta - \delta'|$.*

Assumption 2 is weaker than most of the assumptions in related works. For example, [15] require both $F$ and $1 - F$ to be log-concave. [14] assume that $F$ has $m$-th derivative, and that $\delta - \frac{1 - F(\delta)}{F'(\delta)}$ is greater than some positive constant for all $\delta$, achieving a regret of order $\tilde{\mathcal{O}}(T^{2m+1/4m-1})$. In the case $m = 1$, they propose a different algorithm, reaching a regret $\tilde{\mathcal{O}}(T^{3/4})$. [22] consider Lipschitz-continuous noise, under the additional assumption that, for every $x$, $p^*(x) \in \arg\max_p \pi(x, p)$ is unique, and that $F''$ is bounded. [10] assume quadratic behaviour around every maxima: for every $x$, $p^*(x) \in \arg\max_p \pi(x, p)$, $p^*(x)$ is unique, and for all $p$, $C(p^*(x) - p)^2 \leq \pi(x, p^*(x)) - \pi(x, p) \leq C'(p^*(x) - p)^2$ for some constants $C, C'$. The only work considering non-Lipschitz c.d.f. is [31]; however, they achieve a higher regret bound of $\tilde{\mathcal{O}}(T^{3/4})$.

## 2.3   Information Sharing in Contextual Dynamic Pricing

For $\delta \in \mathbb{R}$, we denote $D(\delta) = \mathbb{P}(\xi_t \geq \delta) = 1 - F(\delta)$, the demand function associated with the noise $\xi_t$. Note that, under Assumption 2, $D$ is $L_\xi$-Lipschitz continuous. Straightforward computations show that, for any *price increment* $\delta \in \mathbb{R}$, the expected reward corresponding to the price $p = g(x_t) + \delta$ in the context $x_t$ is given by

$$\pi(x_t, g(x_t) + \delta) = (g(x_t) + \delta)D(\delta). \tag{2}$$

Equation (2) highlights the intricate roles played by the expected valuation $g(x_t)$ and the price increment $\delta = p - g(x_t)$ in the reward. An immediate consequence is that the optimal price increment $\delta$ depends on the value of $g(x_t)$. Intuitively, if $g(x_t)$ is large, the seller should choose $\delta$ to be small to ensure a high probability $D(\delta)$ to perform a trade. However, for smaller values of $g(x_t)$, the seller might prefer a larger $\delta$ to ensure significant rewards when a trade occurs. Importantly, there is no explicit relationship between the optimal increments $\delta$ for different valuations $g(x_t)$, so knowing the optimal price for a value $g(x_t)$ does not allow optimal pricing for a different value $g(x_{t'})$.

This reasoning suggests that the optimal price increment may span a wide range of values as the expected valuation $g(x_t)$ varies. Unfortunately, as is typical in bandit problems, it is necessary to estimate the reward function around the optimal price with high precision to ensure low regret. Consequently, solving the dynamic pricing problem may entail estimating the demand function precisely across a broad range of price increments. This marks a significant departure from non-contextual dynamic pricing and non-parametric bandit problems, where precise estimation of the reward function is often only necessary around its (single) maximum. Thus, the contextual dynamic pricing problem might be more challenging than its non-contextual counterpart, potentially leading to higher regret. This intuition is supported by the fact that straightforward application of basic bandit algorithms, even in the most simple linear model, leads to regret higher than the rate of order $\tilde{\mathcal{O}}(T^{2/3})$ encountered in non-contextual dynamic pricing problems, as we show in the following discussion.

**Naïve bandit algorithms for contextual dynamic pricing.**   As a first attempt, one might apply a simple explore-then-commit algorithm. Such algorithms start with an exploration phase to obtain uniformly good estimates of both $g$ and of the demand function $D$ over a finite grid of price increments $\{\delta_k\}_{k \in \mathcal{K}}$. Then, in a second exploitation phase, prices are set greedily to maximize the estimated reward. To bound the regret of this approach, note that uniform estimation of $D$ over the grid $\{\delta_k\}_{k \in \mathcal{K}}$ with precision $\epsilon$ requires $\epsilon^{-2}|\mathcal{K}|$ estimation rounds. Moreover, the Lipschitz continuity of the reward function implies a discretization error of order $1/|\mathcal{K}|$. Classical arguments suggest that the regret would be at least $T(\epsilon + 1/|\mathcal{K}|) + |\mathcal{K}|\epsilon^{-2}$, which is minimized for $\epsilon = 1/|\mathcal{K}| = T^{-1/4}$. Thus, this approach would lead to a regret of order $\tilde{\mathcal{O}}(T^{3/4})$.

Another approach, akin to that used in [10], involves partitioning the covariate space into bins and running independent algorithms for non-parametric bandits (such as CAB1 [17]) within each bin. Let us assume, for simplicity, contexts in $[0, 1]$, and that we partition this segment into $K$ bins. Then, the discretization error is $1/K$. Classical results show that the regret in one bin is $\tilde{\mathcal{O}}(T_K^{2/3})$, where

$T_K = T/K$ is the number of rounds in each bin. Consequently, the regret is $\tilde{\mathcal{O}}(T/K + K \times (T/K)^{2/3})$, which is minimized for $K = T^{1/4}$, resulting in a regret $\tilde{\mathcal{O}}(T^{3/4})$.

Thus, both approaches – using either independent bandit algorithms over binned contexts or common exploration rounds followed by an exploitation phase – suffer a regret of order $T^{3/4}$ in the linear model. This raises the question of whether this rate is optimal for the linear model, and if the contextual dynamic pricing problem is indeed more difficult than the non-contextual one. Strikingly, we show that this is not the case. We rely on an intermediate approach, based on regret-minimizing algorithms for each valuation level $g(x_t)$ that *share information* across different values of $g(x_t)$. We show that it achieves an optimal regret rate of order $\tilde{\mathcal{O}}(T^{2/3})$ in the linear valuation model. Moreover, it achieves a rate of order $\tilde{\mathcal{O}}(T^{d+2\beta/d+3\beta})$ in the non-parametric valuation model under minimal assumptions.

## 3 Algorithmic Approach

In this section, we present the general algorithmic approach that we use to tackle dynamic pricing with covariates, called VALUATION APPROXIMATION - PRICE ELIMINATION (VAPE). Before presenting the full scheme, described in Algorithm 1, we start with some intuition that leads to its design. Then, we provide a first analysis of the regret of this algorithm.

### 3.1 Outline of the Algorithm

Equation (2) highlights how the reward is influenced by the expected valuation $g(x_t)$ and by the demand at the price increment $\delta = p_t - g(x_t)$. To separate the effect of these terms, we estimate $g$ and $D$ independently. Hereafter, we assume that the valuations $y_t$ are bounded, in $[-B_y, B_y]$.

**Estimation of $g$.** To estimate $g(x_t)$, we rely on the following observation: when prices $p_t$ are uniformly chosen from the interval $[-B_y, B_y]$, the random variable $2B_y (o_t - 1/2)$ can serve as an unbiased estimate of $g(x_t)$ conditioned on $x_t$. Given that $2B_y (o_t - 1/2)$ is bounded, classical concentration results can be employed to bound the error of our estimates for $g(x_t)$. Thus, in each round, we test whether our estimate of $g(x_t)$ is precise enough to ensure that the error $g(x_t) - \widehat{g}(x_t)$ is small. If this is not the case, we conduct a VALUATION APPROXIMATION round by setting a uniform price. In the next sections, we consider linear and non-parametric valuation functions, and we discuss how to ensure sufficient precision in a limited number of valuation approximation rounds.

Previous approaches for estimating valuation functions in the linear model include the regularized maximum-likelihood estimator [15, 30], which requires knowledge of the noise distribution. Another approach used in [22] relies on the relation between estimating a linear valuation function from binary feedback and the classical linear classification problem. The authors propose recovering the linear parameters $\theta$ through logistic regression; however, they do not provide an explicit estimation rate for $\theta$. [20] use the EXP-4 algorithm to aggregate policies corresponding to different values of $\theta$ and $F$, thus circumventing the necessity to estimate them. In a similar vein, in the non-parametric valuation model, [10] avoid the need to estimate $g(x_t)$ by employing independent bandit algorithms for each (binned) value of $x_t$. Closer to our method are the works of [14] and [21], who also set uniform prices to obtain unbiased estimates of the valuations. Nonetheless, their algorithms are significantly different from ours. First, they propose two-phased algorithms for which the phase length is set beforehand. Such an approach necessitates additional assumptions on how contexts are drawn; specifically, contexts are assumed to be i.i.d. from a distribution with a lower bound on the eigenvalues of the covariance matrix. This is needed to ensure that contexts observed in the first phase can represent the context distribution well. By contrast, our phases are adaptive, allowing our algorithm also to handle adversarial contexts and render these assumptions superfluous. Second, we obtain better regret rates by using piecewise-constant estimators, fitted in a regret-minimization sub-routine, as detailed in the next paragraph. On the other hand, [14] performs a phase of pure exploitation, relying on an estimate of the CDF $F$ that is constructed using Kernel methods. [21], instead, re-frames the problem as a perturbed linear bandit, which exhibits a regret linear in the dimension. However, this dimension depends on the size of the discretization grid – which is horizon dependent – leading to worse rates.

**Estimation of $D$.** If the expected valuation $g(x_t)$ is known with sufficient precision, we can use it to estimate the demand function over a set of candidate price increments $\{\delta_k\}_{k \in \mathcal{K}}$. More precisely, assume we set a price $p_t = \widehat{g}(x_t) + \delta_k$, and that $|\widehat{g}(x_t) - g(x_t)| \le \epsilon$. Then, the observation $o_t$ can

---

**Algorithm 1** VALUATION APPROXIMATION - PRICE ELIMINATION (VAPE): General scheme

---
1: **Input**: Price increments $\{\delta_k\}_{k\in\mathcal{K}}$, expected valuation precision $\text{err}_t(x)$, reward confidence intervals $[\text{LCB}_t(k), \text{UCB}_t(k)]$, parameters $\alpha, \epsilon$.
2: **while** $t \leq T$ **do**
3:      **if** $\text{err}_t(x_t) > \epsilon$ **then**                             ▷ Valuation Approximation
4:          Post a price $p_t \sim \mathcal{U}([-B_y, B_y])$
5:          Use $o_t$ to improve the valuation estimator $\widehat{g}(x_t)$
6:      **else**                                                 ▷ Price Elimination
7:          $\mathcal{A}_t \leftarrow \{k \in \mathcal{K} : \widehat{g}_t + \delta_k \in [0, B_y]\}$
8:          $\mathcal{K}_t \leftarrow \{k \in \mathcal{A}_t : \text{UCB}_t(k) \geq \max_{k' \in \mathcal{A}_t} \text{LCB}_t(k')\}$
9:          Choose $k_t \in \arg\min_{k \in \mathcal{K}_t} N_t^k$ and post a price $p_t = \widehat{g}_t + \delta_{k_t}$
10:          Update $\widehat{D}_{t+1}^{k_t}, N_{t+1}^{k_t}$

---

be used as an almost unbiased estimate of the demand at level $\delta_k$, since

$$\mathbb{E}[o_t] = \mathbb{E}\left[\mathbb{1}\{\widehat{g}(x_t) + \delta_k \leq g(x_t) + \xi_t\}\right] = D(\delta_k + \widehat{g}(x_t) - g(x_t)).$$

Under Assumption 2, $D$ is $L_\xi$-Lipschitz, so the bias is of order $L_\xi\epsilon$. Then, relying on classical bandit techniques, we show that with high probability (for $\alpha$ small enough), $|D(\delta_k) - \widehat{D}_t^k|$ is of order $L_\xi\epsilon + \sqrt{\log(1/\alpha)/N_t^k}$, where $\widehat{D}_t^k$ is the average of the observations $o_t$ when setting a price $p_t = \widehat{g}(x_t) + \delta_k$, and $N_t^k$ is the number of rounds in which we chose the price increment $\delta_k$ up to round $t$. Importantly, to estimate $\widehat{D}_t^k$, we share information collected during *all rounds* we chose the increment $\delta_k$ across all values of $\widehat{g}(x_t)$; this is necessary to obtain better regret rates. Then, using $p_t\widehat{D}_t^k$ as an estimate of the reward $\pi(x_t, p_t)$ given the price $p_t = \widehat{g}(x_t) + \delta_k$, the error $|\pi(x_t, p_t) - p_t\widehat{D}_t^k|$ is of order $B_y(L_\xi\epsilon + \sqrt{\log(1/\alpha)/N_t^k})$.

The PRICE ELIMINATION subroutine relies on the previous remark to select a price increment. For each increment $\delta_k$, we build a confidence bound $[\text{LCB}_t(\delta_k), \text{UCB}_t(\delta_k)] = [p_t\widehat{D}_t^k \pm B_y(2L_\xi\epsilon + \sqrt{2\log(1/\alpha)/N_t^k})]$ for the reward of price $p_t = \widehat{g}(x_t) + \delta_k$. Then, we use a successive elimination algorithm [13, 25] to select a good increment. More precisely, we consider increments $\delta_k$ such that $\text{UCB}_t(\delta_k) \geq \max_l \text{LCB}_t(\delta_l)$, and we choose among these increments the increment $\delta_{k_t}$ that has been selected the least frequently. By doing so, we ensure to only select potentially optimal prices and gradually eliminate sub-optimal increments.

## 3.2 A First Bound on the Regret

Before discussing the application of the algorithmic scheme VAPE to linear and non-parametric valuation functions, we provide some intuition on regret bounds achievable through this scheme.

**Claim 1.** (Informal) *Let* $\delta_k = k\epsilon$ *for* $k \in \mathcal{K} \triangleq [\![\lfloor -B_y-1/\epsilon\rfloor, \lceil B_y+1/\epsilon\rceil]\!]$. *Assume that, on a high-probability event,* $|\widehat{g}(x_t) - g(x_t)| \leq \epsilon$ *for every round $t$ where* PRICE ELIMINATION *is conducted. Then, on a high-probability event, the regret of* VAPE *verifies*

$$R_T \lesssim T^{\text{VA}}(\epsilon) + T\epsilon + \log(1/\alpha)\epsilon^{-2}.$$

*where* $T^{\text{VA}}(\epsilon)$ *is a bound on the length of the* VALUATION APPROXIMATION *phase.*

Claim 1 is proved in the Appendix by combining Equations (4) and (5), and Lemma 4. We provide a sketch of proof below. To bound on regret of VAPE using Claim 1, it will suffice to bound the length of the VALUATION APPROXIMATION phase, and prove high-probability error bounds on $g(x_t)$.

*Sketch of proof.* Note that the regret in the VALUATION APPROXIMATION phase scales at most linearly with its length. Then, to prove Claim 1, it is enough to bound the regret during the PRICE ELIMINATION phase. We begin by bounding the sub-optimality gap of the price chosen at round $t$, showing that it is of order $\epsilon + \sqrt{\log(1/\alpha)/N_t^{k_t}}$.

To do so, for $p \in \mathbb{R}$, we define $\Delta_t(x_t, p) = \max_{p'} \pi(x_t, p') - \pi(x_t, p)$ the sub-optimality gap corresponding to price $p$. Recall that $\delta_{k_t}$ is the increment chosen at round $t$, i.e. that $p_t = \widehat{g}(x_t) + \delta_{k_t}$.

Classical arguments from the bandit literature show that with high probability, for all $k \in \mathcal{K}$, the upper and lower confidence bounds on $\pi(x_t, \widehat{g}(x_t) + \delta_k)$ given by $\text{UCB}_t(\delta_k)$ and $\text{LCB}_t(\delta_k)$ are valid. Then, the optimal increment $\delta_{k^*}$ defined by $k^* = \arg\max_{k \in \mathcal{A}_t} \pi(x_t, \widehat{g}(x_t) + \delta_k)$ belongs to the set of non-eliminated increments. Now, on the one hand, since $\text{UCB}_t(\delta_{k_t}) \geq \text{LCB}_t(\delta_{k_t^*})$, and since the confidence interval are valid, the gap $\pi(x_t, \widehat{g}(x_t) + \delta_{k_t^*}) - \pi(x_t, p_t)$ is of order $\epsilon + \sqrt{2 \log(1/\alpha)/N_t^{k_t}} + \sqrt{2 \log(1/\alpha)/N_t^{k_t^*}}$. Our round-robin sampling scheme ensures that $N_t^{k_t^*} \geq N_t^{k_t}$, so this bound is of order $\epsilon + \sqrt{\log(1/\alpha)/N_t^{k_t}}$. On the other hand, our choice of grid $\{\delta_k\}_{k \in \mathcal{K}}$, together with the Lipschitz-continuity of the reward in Assumption 2, imply that the cost $\Delta_t(x_t, \widehat{g}(x_t) + \delta_{k_t^*})$ of considering a discrete price grid is of order $B_y L_\xi \epsilon$. Thus, at each round, the gap $\Delta_t(x_t, \widehat{g}(x_t) + \delta_{k_t})$ is at most of order $\epsilon + \sqrt{\log(1/\alpha)/N_t^{k_t}}$ (up to problem-dependent constants).

Now, let us decompose the regret of the PRICE ELIMINATION phase as follows:

$$\sum_{t \in \text{PRICE ELIMINATION phase}} \Delta(x_t, p_t) = \sum_{k \in \mathcal{K}} \sum_{t: k_t = k} \Delta(x_t, p_t).$$

In order to bound $\sum_{t: k_t = k} \Delta(x_t, p_t)$ for $k \in \mathcal{K}$, we begin by introducing further notations. Let us denote $\tau_1^k, \ldots, \tau_T^k$ the rounds in the PRICE ELIMINATION phase where we choose $k_t = k$. We also define $\Delta_a = 2^{-a}$ and $\overline{a}$ such that $\Delta_{\overline{a}} \approx \epsilon$. For all $a \leq \overline{a}$, we also define $t_a$ such that the bound $\epsilon + \sqrt{\log(1/\alpha)/t_a}$ is of order $\Delta_a$. Then, our previous reasoning implies that if $i \geq t_a$ for some $a \in \{1, \overline{a}\}$, it must be that $\Delta_t(x_t, p_{\tau_i^k}) \leq \Delta_a$. Moreover, for $a \geq 1$, each phase $\{t_a, \ldots, t_{a+1}\}$ is of length approximately $\log(1/\alpha)(\Delta_{a+1}^{-2} - \Delta_a^{-2})$. Thus,

$$\sum_{t: k_t = k} \Delta(x_t, p_t) \lesssim \frac{\log(1/\alpha)}{\Delta_1} + \sum_{a=1}^{\overline{a}-1} \Delta_a \times \left( \frac{\log(1/\alpha)}{\Delta_{a+1}^2} - \frac{\log(1/\alpha)}{\Delta_a^2} \right) + \Delta_{\overline{a}} N_T^k.$$

Using the definitions of $\Delta_a$ and $\overline{a}$, we find that this sum is of order $\log(1/\alpha)/\epsilon + \epsilon N_T^k$. We conclude by summing over the values of $k \in \mathcal{K}$, using $\sum_{k \in \mathcal{K}} N_T^k \leq T$ and the fact that $|\mathcal{K}|$ is of order $\epsilon^{-1}$. $\square$

# 4 Linear Valuation Functions

In this section, we consider the linear valuation model, given by

$$g(x) = x^\top \theta, \tag{3}$$

where $\theta \in \mathbb{R}^d$ is an unknown parameter. To ensure that the valuations are bounded, we assume the boundedness of the parameter $\theta$.

**Assumption 3.** *The parameter $\theta$ is bounded: $\|\theta\| \leq B_\theta$*

Note that under Assumptions 1 and 3, the expected valuations $g(x_t)$ verify $|g(x_t)| \leq B_g$ for $B_g = B_x \times B_\theta$. Moreover, the random valuations verify a.s. $|y_t| \leq B_y$ for $B_y = B_g + B_\xi$.

We apply the VAPE algorithmic scheme to the problem of dynamic pricing with linear valuations. To estimate the valuation function, we use a ridge estimator for the parameter $\theta$. Moreover, we distinguish between phases by setting $\iota_t = 1$ if $t$ belongs to the VALUATION APPROXIMATION phase and $\iota_t = 0$ if $t$ belongs to the PRICE ELIMINATION one. The details are presented in Algorithm 2.

**Theorem 1.** *Assume that the valuations follow the model given by Equations* (1) *and* (3). *Under Assumptions 1, 2, and 3, the regret of Algorithm* VAPE *for Linear Valuations with parameters* $\epsilon = \left( d^2 \log(T)^2 / T \right)^{1/3}$, $\mu = \epsilon / \left( B_y \sqrt{d \log\left( \frac{1+B_x^2 T}{\alpha} \right)} + B_\theta \right)$, *and* $\alpha = 1/\left( T + 2T^2 \left( 3 + (B_\xi + 1) T^{1/3} \right) \right)$ *verifies*

$$R_T \leq C_{B_\xi, B_x, B_\theta, L_\xi} d^{2/3} T^{2/3} \log(T)^{2/3}$$

*with probability $1 - T^{-1}$, where $C_{B_\xi, B_x, B_\theta, L_\xi}$ is a constant that polynomially depends on $B_\xi$, $B_x$, $B_\theta$, and $L_\xi$.*

*Sketch of proof.* [See Appendix B for the full proof] Using Claim 1, we see that it is enough to prove that the VALUATION APPROXIMATION phase allows to estimate $g(x_t)$ up to precision $\epsilon = \left( d^2 \log(T)^2 / T \right)^{1/3}$ in at most $O(d^{2/3} T^{2/3} \log(T)^{2/3})$ rounds.

**Algorithm 2** VALUATION APPROXIMATION - PRICE ELIMINATION (VAPE) for Linear Valuations

1: **Input**: bounds $B_y$ and $L_\xi$, parameters $\alpha$, $\mu$, $\epsilon$.
2: **Initialize**: $\widehat{\theta}_1 = \mathbf{0}_d$, $\mathbf{V}_1 = \mathbf{I}_d$, $K = \lceil (B_y+1)/\epsilon \rceil$, $\mathcal{K} = [\![-K, K]\!]$, and for $k \in \mathcal{K}$, $N_1^k = \widehat{D}_1^k = 0$.
3: **while** $t \leq T$ **do**
4:      **if** $\|x_t\|_{\mathbf{V}_t^{-1}} > \mu$ **then**                                  ▷ Valuation Approximation
5:          Post a price $p_t \sim \mathcal{U}([-B_y, B_y])$
6:          $\iota_t \leftarrow 1$, $\mathbf{V}_{t+1} \leftarrow \sum_{s \leq t} \iota_s x_s x_s^\top + \mathbf{I}_d$, $\widehat{\theta}_{t+1} \leftarrow 2B_y \mathbf{V}_{t+1}^{-1} \sum_{s \leq t} \iota_s \left(o_s - \frac{1}{2}\right) x_s$
7:      **else**                                                         ▷ Price Elimination
8:          $\iota_t \leftarrow 0$, $\widehat{g}_t \leftarrow x_t^\top \widehat{\theta}_t$, $\mathcal{A}_t \leftarrow \{k \in \mathcal{K} : \widehat{g}_t + k\epsilon \in [0, B_y]\}$
9:          **for** $k \in \mathcal{A}_t$ **do**
10:              $\mathrm{UCB}_t(k) \leftarrow (\widehat{g}_t + k\epsilon)\left(\widehat{D}_t^k + \sqrt{\frac{2\log(1/\alpha)}{N_t^k}} + 2L_\xi\epsilon\right)$
11:              $\mathrm{LCB}_t(k) \leftarrow (\widehat{g}_t + k\epsilon)\left(\widehat{D}_t^k - \sqrt{\frac{2\log(1/\alpha)}{N_t^k}} - 2L_\xi\epsilon\right)$
12:          $\mathcal{K}_t \leftarrow \{k \in \mathcal{A}_t : \mathrm{UCB}_t(k) \geq \max_{k' \in \mathcal{A}_t} \mathrm{LCB}_t(k')\}$
13:          Choose $k_t \in \arg\min_{k \in \mathcal{K}_t} N_t^k$ and post a price $p_t = \widehat{g}_t + k_t\epsilon$
14:          Update $\widehat{D}_{t+1}^{k_t} \leftarrow \frac{N_t^{k_t}\widehat{D}_t^{k_t} + o_t}{N_t^{k_t} + 1}$, $N_{t+1}^{k_t} \leftarrow N_t^{k_t} + 1$.

To prove the first part of the claim, note that for all rounds in the PRICE ELIMINATION phase, $\|x_t\|_{\mathbf{V}_t^{-1}} \leq \mu = \epsilon/\left(B_y\sqrt{d\log\left(1+B_x^2 T/\alpha\right)}+B_\theta\right)$. Then,

$$|\widehat{g}(x_t) - g(x_t)| \leq \|\theta - \widehat{\theta}_t\|_{\mathbf{V}_t}\|x_t\|_{\mathbf{V}_t^{-1}} \leq \|\theta - \widehat{\theta}_t\|_{\mathbf{V}_t} \times \epsilon/\left(B_y\sqrt{d\log\left(1+B_x^2 T/\alpha\right)}+B_\theta\right).$$

Classical result on ridge regression in bandit framework [1] show that on a large probability event, $\|\theta - \widehat{\theta}_t\|_{\mathbf{V}_t} \leq \left(B_y\sqrt{d\log\left(1+B_x^2 T/\alpha\right)} + B_\theta\right)$, so $|\widehat{g}(x_t) - g(x_t)| \leq \epsilon$.

To prove the second part of the claim, we rely on the elliptical potential lemma to bound the number of rounds where $\|x_t\|_{\mathbf{V}_t^{-1}} \geq \mu$. This Lemma states that $\sum_{i=1}^{|\mathcal{G}|} \|x_{t_i}\|_{\mathbf{V}_{t_i-1}^{-1}} \leq \sqrt{|\mathcal{G}|d\log\left(|\mathcal{G}|+d/d\right)}$, where $t_i$ is the $i$-th round of the VALUATION APPROXIMATION phase, and $|\mathcal{G}|$ is its length. Using the fact that $\|x_{t_i}\|_{\mathbf{V}_{t_i-1}^{-1}} \geq \mu$, we conclude that $|\mathcal{G}| \leq \frac{d\log(T+d/d)}{\mu^2}$, which implies the result.    □

Theorem 1 provides a regret bound of order $\tilde{O}(T^{2/3})$, showing that VAPE for Linear Valuations is minimax optimal, possibly up to sub-logarithmic terms and to sub-linear dependence in the dimension. Indeed, it matches the $T^{2/3}$ lower bound established in [31] for linear valuation functions and Lipschitz-continuous demand functions. This result represents a clear improvement over the existing regret bounds for the same problem. Indeed, VAPE achieves the regret bound conjectured in [22] while at the same time removing their regularity assumption on the revenue function. On the other hand, we improve on the regret rate $\tilde{\mathcal{O}}(T^{3/4})$ achieved respectively in [31] under assumptions slightly milder than ours, and in [14] under stronger assumptions.

## 5 Non-Parametric Valuation Functions

In this Section, we consider the non-parametric valuation model. As usual in dynamic pricing, we assume that the valuation function $g$ is bounded. Furthermore, we assume that it is $(L_g, \beta)$-Hölder continuous for some constants $L_g > 0$ and $0 < \beta \leq 1$.

**Assumption 4.** *The valuation function $g$ is $(L_g, \beta)$-Hölder: for all $(x, x') \in \mathbb{R}^d$, $|g(x) - g(x')| \leq L_g \|x - x'\|^\beta$.*

Under Assumptions 1 and 2, the random valuations $y_t$ verify $|y_t| \leq B_y$ for $B_y = B_\xi + B_g$.

Next, we apply the VAPE algorithmic scheme to the non-parametric valuation model. To estimate the function g, we use a finite grid of points, on which this function is evaluated. More precisely, we

**Algorithm 3** VALUATION APPROXIMATION - PRICE ELIMINATION (VAPE) for Non-Parametric Valuations

1: **Input**: bounds $B_y$ and $L_\xi$, finite set $\overline{\mathcal{X}} \subset \mathbb{R}^d$, parameters $\alpha, \tau, \epsilon$.
2: **Initialize**: $\mathcal{G}_{\overline{x}} = \emptyset$ for all $\overline{x} \in \overline{\mathcal{X}}$, $K = \lceil B_y + 1/\epsilon \rceil$, $\mathcal{K} = [\![-K, K]\!]$, and for $k \in \mathcal{K}$, $N_1^k = \widehat{D}_1^k = 0$.
3: **while** $t \le T$ **do** $\overline{x}_t \leftarrow \arg\min_{\overline{x}' \in \overline{\mathcal{X}}} \|x_t - \overline{x}'\|$
4:     **if** $|\mathcal{G}_{\overline{x}_t}| < \tau$ **then**                                        ▷ Price Elimination
5:         Post a price $p_t \sim \mathcal{U}([-B_y, B_y])$
6:         $\mathcal{G}_{\overline{x}_t} \leftarrow \mathcal{G}_{\overline{x}_t} \cup \{t\}, \widehat{g}(\overline{x}_t) \leftarrow \frac{2B_y}{|\mathcal{G}_{\overline{x}_t}|} \sum_{s \in \mathcal{G}_{\overline{x}_t}} \left(o_s - \frac{1}{2}\right)$
7:     **else**                                                         ▷ Run Successive Elimination
8:         $\widehat{g}_t \leftarrow \widehat{g}(\overline{x}_t), \mathcal{A}_t \leftarrow \{k \in \mathcal{K} : \widehat{g}_t + k\epsilon \in [0, B_y]\}$
9:         **for** $k \in \mathcal{A}_t$ **do**
10:             $\text{UCB}_t(k) \leftarrow (\widehat{g}_t + k\epsilon)\left(\widehat{D}_t^k + \sqrt{\frac{2\log(1/\alpha)}{N_t^k}} + 2L_\xi\epsilon\right)$
11:             $\text{LCB}_t(k) \leftarrow (\widehat{g}_t + k\epsilon)\left(\widehat{D}_t^k - \sqrt{\frac{2\log(1/\alpha)}{N_t^k}} - 2L_\xi\epsilon\right)$
12:         $\mathcal{K}_t \leftarrow \{k \in \mathcal{A}_t : \text{UCB}_t(k) \ge \max_{k' \in \mathcal{A}_t} \text{LCB}_t(k')\}$
13:         Choose $k_t \in \arg\min_{k \in \mathcal{K}_t} N_t^k$ and post a price $p_t = \widehat{g}_t + k_t\epsilon$
14:         Update $\widehat{D}_{t+1}^{k_t} \leftarrow \frac{N_t^{k_t}\widehat{D}_t^{k_t} + o_t}{N_t^{k_t} + 1}, N_{t+1}^{k_t} \leftarrow N_t^{k_t} + 1$.

consider a minimal $(\epsilon/3L_g)^{1/\beta}$-covering $\overline{\mathcal{X}}$ of the ball of radius $B_x$ in $R^d$, i.e. a finite set of points, of minimal cardinality, such that for any context $x$ such that $\|x\| \le B_x$, there exists a point in $\overline{\mathcal{X}}$ at a distance at most $(\epsilon/3L_g)^{1/\beta}$ from $x$.

At each round, we round the context $x_t$ to the closest context $\overline{x}$ in $\overline{\mathcal{X}}$ by setting $\overline{x}_t = \arg\min_{\overline{x}' \in \overline{\mathcal{X}}} \|x_t - \overline{x}'\|$, and acting as if we observed the context $\overline{x}_t$. If this context has not been observed sufficiently, we conduct a round of VALUATION APPROXIMATION: we sample a price uniformly at random and use it to update our estimate of $g(\overline{x}_t)$; otherwise, we proceed with the PRICE ELIMINATION phase. To distinguish between the VALUATION APPROXIMATION steps corresponding to contexts $\overline{x} \in \overline{\mathcal{X}}$, we collect their indices in sets $\mathcal{G}_{\overline{x}}$. The algorithm is presented in Algorithm 3.

**Theorem 2.** *Assume that the valuations follow the model given by Equation* (1). *Under Assumptions 1, 2, and 4, with probability* $1 - T^{-1}$ *the regret of Algorithm* VAPE *for non-parametric Valuations with parameters* $\epsilon = (T/\log(T))^{\frac{-\beta}{d+3\beta}}$, $\alpha = T^{-4}$, $\tau = {}^{18B_y^2}\log(2|\overline{\mathcal{X}}|/\alpha)/\epsilon^2$, *and* $\overline{\mathcal{X}}$ *a minimal* $(\epsilon/3L_g)^{1/\beta}$-*covering of the ball of radius* $B_x$ *verifies*

$$R_T \le C_{B_x, B_g, B_\xi, L_g, L_\xi, d, \beta} T^{\frac{d+2\beta}{d+3\beta}} \log(T)^{\frac{\beta}{d+3\beta}},$$

*where* $C_{B_x, B_g, B_\xi, L_g, L_\xi, d, \beta}$ *is a constant that polynomially depends on* $B_x$, $B_g$, $B_\xi$, $L_g$, $L_\xi$, $d$, *and* $\beta$.

*Sketch of proof.* [See Appendix C for the full proof] Using Claim 1, we only need to show that the length of the VALUATION APPROXIMATION phase is at most of order $T^{d+2\beta/d+3\beta} \log(T)^{\beta/d+3\beta}$ and that w.h.p., it allows estimating $g$ uniformly on a ball of radius $B_x$ with precision $\epsilon = (T/\log(T))^{-\beta/d+3\beta}$.

To prove the first part of the claim, we note that classical results imply that the size of a minimal covering of precision $\epsilon^{1/\beta}$ of a ball in dimension $d$ scales as $\epsilon^{-d/\beta}$. Then, the total length of the VALUATION APPROXIMATION phase is of order $\epsilon^{-d/\beta}\tau \approx T^{d+2\beta/d+3\beta} \log(T)^{\beta/d+3\beta}$. To prove the second part of the lemma, note that the Hölder-continuity of $g$ and the definition of the $(\epsilon/3L_g)^{1/\beta}$-covering $\mathcal{G}$ ensure that $|g(x_t) - g(\overline{x}_t)| \le \epsilon/3$. Then, standard concentration arguments reveal that $\tau \approx \log(|\overline{\mathcal{X}}|/\alpha)/\epsilon^2$ samples are sufficient to estimate $g(\overline{x}_t)$ with precision $\epsilon$ with high probability. $\quad\square$

Theorem 2 shows that the Algorithm VALUATION APPROXIMATION – PRICE ELIMINATION for non-parametric valuations enjoys a $\tilde{O}(T^{d+2\beta/d+3\beta})$ regret bound when the noise c.d.f. is Lipschitz and the valuation function Hölder-continuous. This result is the first of its kind under such minimal assumptions. In particular, previous work by [10] assumes quadratic behavior around the optimal price for all values of $g(x)$ – a very strong assumption. However, this rate is higher than the $\tilde{O}(T^{d+\beta/d+2\beta})$

rates that are usually encountered in $\beta$-Hölder non-parametric bandits [7]. Thus, the question of optimality of the VAPE algorithmic scheme in the non-parametric valuation problem remains open.

## 6 Conclusions

In this paper, we studied the problem of dynamic pricing with covariates. We first presented a novel algorithmic approach called VAPE, which adaptively alternates between improving the valuation approximation and learning to set prices through successive elimination. We then applied VAPE under two valuation models – when the buyer's valuation corresponds to a noisy linear function and when expected valuations follow a smooth non-parametric model. In the linear case, our regret bounds are order-optimal, while in the non-parametric setting, we improve existing results. All our results are proven under regularity assumptions that are either milder or match existing assumptions.

Our results on the linear valuation model are the first to match the existing lower bound rate of $\Omega(T^{2/3})$ under our assumptions. However, the optimal dependence of this rate on the dimension of the context remains unknown. Additionally, there are no similar lower bounds for non-parametric valuations. We conjecture that our results are also tight in this setting but leave this for future work. Future research directions also include exploring other valuation models, and further relaxing our assumptions, as Lipschitz-continuity of the noise (Assumption 2). Without this, even minor increases in the price could lead to a major drop in revenue, magnifying the impact of valuation approximation errors. Another limiting assumption is that the noise is independent and identically distributed, such that its distribution can be learned across different contexts. It is of great interest to study problems where the noise distribution can change between rounds, or depends on the context.

## Broader Impacts

As all pricing problems, dynamic pricing can have both positive and negative impacts – offering prices that are more suited to the buyers on the one hand, while increasing the seller's revenue at the expense of buyers on the other hand. In addition, as with many contextual problems, there might be biases and challenges involving fairness – one should make sure that similar customers are offered similar prices. While acknowledging these issues, our work was meant to focus only on the theoretical analysis of what is considered a well-established problem in literature, leaving the study of these related topics as future work.

## Acknowledgments

This project has received funding from the European Union's Horizon 2020 research and innovation programme under the Marie Skłodowska-Curie grant agreement No 101034255. Solenne Gaucher gratefully acknowledges funding from the Fondation Mathématique Jacques Hadamard. Vianney Perchet acknowledges support from the French National Research Agency (ANR) under grant number (ANR-19-CE23-0026 as well as the support grant, as well as from the grant "Investissements d'Avenir" (LabEx Ecodec/ANR-11-LABX-0047). This research was supported in part by the French National Research Agency (ANR) in the framework of the PEPR IA FOUNDRY project (ANR-23-PEIA-0003) and through the grant DOOM ANR-23-CE23-0002. It was also funded by the European Union (ERC, Ocean, 101071601). Views and opinions expressed are however those of the author(s) only and do not necessarily reflect those of the European Union or the European Research Council Executive Agency. Neither the European Union nor the granting authority can be held responsible for them.

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

# A   Simulations

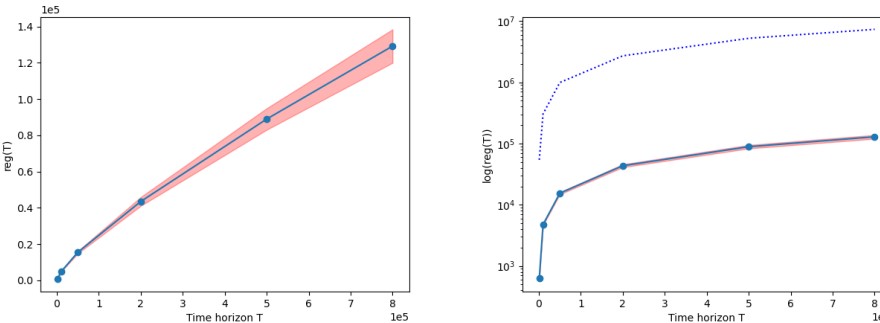

Figure 1: The plots here show the regrets rate of VAPE for linear evaluations, both in the standard and logarithmic scale (left and right respectively). The solid lines represent the average of the performance over 15 repetitions of the routine. The faded red area shows the standard error, while in the right subplot the dotted line corresponds to the theoretical regret bound.

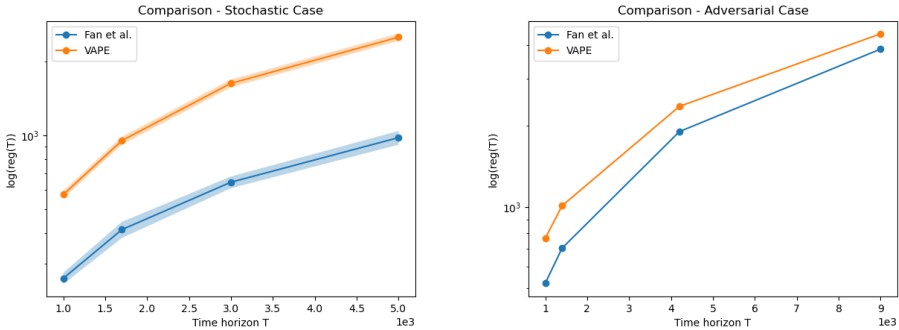

Figure 2: The two subplots show a comparison between VAPE and the algorithm in [14] in the stochastic and adversarial case, where the time horizons used are $T \in [1000, 1700, 3000, 5000]$ (left subplot), and $T \in [1000, 1400, 4200, 9000]$ (right subplot). In both cases the solid lines represent the average of the regret rates across the 15 repetitions of the simulations, while the faded area the standard error. In the subplot on the right, due to the specificity of the setting, the variance across runs is minimal, hence the faded area results invisible. The regret graph is in both cases plotted in logaritmic scale.

In this section, we illustrate some numerical simulations that aim to show the empirical performance of our VAPE algorithm for Linear Valuations. Moreover we present a comparison with the algorithm proposed in [14] in the case in which the only regularity assumed on the CDF of the noise distribution is Lipschitzness. The code implemented for these simulations is publicly available in the repository: `https://github.com/MatildeTulii1/Improved-Algorithms-for-Contextual-Dynamic-Pricing`

**VAPE**   In order to test our algorithm, we built a dataset of 5 contexts belonging to $\mathbb{R}^3$ generated by a canonical gaussian distribution and subsequently normalized. Throughout the run the contexts are chosen from this set uniformly at random, while the noise term is picked from a gaussian distribution truncated between $-1$ and $1$ with mean 0 and variance 0.1. Similarly, also the parameter $\theta$ is a normalized vector initially drawn from a gaussian distribution. The algorithm has been tested on time horizons $T \in [1000, 10000, 50000, 200000, 500000, 800000]$, and the hyperparameters $\alpha, \mu, \epsilon$ are set as in the statement of Theorem 1. Figure 1 shows the results of this implementation. The empirical regret rates of VAPE respect the theoretical upper-bound expressed in the paper, moreover it shows optimal computational times that can handle big time horizons.

**Comparison with [14]**   Next we compare our algorithm with the algorithm proposed in Appendix F of [14], in which they propose a routine to tackle the dynamic pricing problem with linear valuation in the case in which the CDF $F$ is Lipschitz. The comparison is carried out in two different settings: a stochastic and an adversarial one, and to make it more fair both algorithm receive as input the time horizon $T$.

In the stochastic case, similarly as before we consider a set of possible contexts in $\mathbb{R}^3$ drawn uniformly at random and then normalised. During the routine, at each time step one of these is randomly selected. This method of receiving contexts meets the assumption included in [14], making sure that no eigenvalue of the covariance matrix of the distribution of contexts is too small. As for the contexts, the parameter $\theta$ is selected *ex-novo* with every new run of the algorithm. We chose to implement a comparison with this specific algorithms since it was among the closest with our work, as discussed in the main paper, but its prohibitive computational costs make difficult to see the good behaviour of VAPE which, being based on a bandit approach, requires bigger time horizons to converge.

The adversarial case, instead is a toy example which is purposely designed to badly interfere with the algorithm proposed by [14]. In this case the set of contexts is made of only two samples of orthogonal vectors, specifically in the form $[x, 0, z]$ and $[0, 1, 0]$. To make sure that the effect of this choice of contexts is not invalidated by the parameter $\theta$, this is considered to be fixed as $[0.3, 0.3, 0.3]$. The algorithm receives the first context during the exploration phase and the second during the exploitation one, such that the information gathered initially result meaningless in the latter subroutine. As before the computational costs of [14] limited the time horizons on which we were able to run this simulation, still it can be noted how VAPE, exposed to the same contexts in the same order, does not suffer from such choice, since the phases are defined adaptively, thus its regret rates remain consistent with the stochastic case. The results of this comparison are shown in Figure 2.

## B   Proof of Theorem 1

We state several lemmas before proving Theorem 1. We begin by bounding the length of the exploration phase corresponding to lines 5 and 6 of Algorithm 2.

**Lemma 1.** *Let $\mathcal{G} = \{t \leq T : \iota_t = 1\}$. Almost surely, the length of exploration phase $\mathcal{G}$ is bounded as*

$$|\mathcal{G}| \leq \frac{d \log\left(\frac{T+d}{d}\right)}{\mu^2}.$$

The following lemma bounds the error of our estimates for $\theta$ and $D$, for the values of $\mu$ prescribed in Theorem 1. Before stating the Lemma, we define the event

$$\mathcal{E} = \left\{ \forall t \notin \mathcal{G}, |\widehat{g}_t - g(x_t)| \leq \epsilon, \text{and} \left| \widehat{D}_t^k - D(k\epsilon) \right| \leq \sqrt{\frac{2 \log(1/\alpha)}{N_t^k}} + L_\xi \epsilon \right\}.$$

**Lemma 2.** *The event $\mathcal{E}$ happens with probability at least $1 - (\alpha + 2T^2 |\mathcal{K}| \alpha)$.*

Finally, we bound the number of times a sub-optimal price increment $k\epsilon$ can be selected. For $p \in \mathbb{R}$, $x \in \mathbb{R}^d$, we define

$$\Delta(x, p) = \sup_{p' \in [0, B_y]} \pi(x, p') - \pi(x, p).$$

**Lemma 3.** *On the event $\mathcal{E}$, for all $t \notin \mathcal{G}$, if $k_t = k$, then $k$ must be such that*

$$\Delta(x_t, \widehat{g}_t + k\epsilon) \leq B_y \left( 4\sqrt{\frac{2 \log(1/\alpha)}{N_t^k}} + 9L_\xi \epsilon \right).$$

We are now ready to bound the regret of Algorithm VAPE for Linear Valuations. We begin by rewriting the regret as

$$R_T = \sum_{t=1}^{T} \left( \max_{p \in [0, B_y]} \pi(x_t, p) - \pi(x_t, p_t) \right)$$

$$= \sum_{t \in \mathcal{G}} \left( \max_{p \in [0, B_y]} \pi(x_t, p) - \pi(x_t, p_t) \right) + \sum_{t \notin \mathcal{G}} \left( \max_{p \in [0, B_y]} \pi(x_t, p) - \pi(x_t, p_t) \right). \tag{4}$$

Under Assumptions 1, 2, and 3, both the optimal price and $p_t$ are in $[0, B_y]$, we know that the instantaneous regret is bounded by $B_y$. Then,

$$\sum_{t \in \mathcal{G}} \left( \max_{p \in [0, B_y]} \pi(x_t, p) - \pi(x_t, p_t) \right) \le B_y |\mathcal{G}|. \tag{5}$$

Using Lemma 1 together with the definition of $\mu$, we find that

$$\sum_{t \in \mathcal{G}} \max_{p \in [0, B_y]} \left( \pi(x_t, p) - \pi(x_t, p_t) \right) \le \frac{B_y d \log\left(\frac{T+d}{d}\right) \left( B_y \sqrt{d \log\left(\frac{B_x T+1}{\alpha}\right)} + B_\theta \right)^2}{\epsilon^2}. \tag{6}$$

We rely on the following Lemma to bound $\sum_{t \notin \mathcal{G}} \left( \max_{p \in [0, B_y]} \pi(x_t, p) - \pi(x_t, p_t) \right)$.

**Lemma 4.** *On the event $\mathcal{E}$,*

$$\sum_{t \notin \mathcal{G}} \left( \max_{p \in [0, B_y]} \pi(x_t, p) - \pi(x_t, p_t) \right) \le |\mathcal{K}| \left( 512 B_y \log(1/\alpha) + 22 \frac{B_y \log(1/\alpha)}{L_\xi \epsilon} \right) + 36 B_y T L_\xi \epsilon.$$

Combining Equations (4), (6), and Lemma 4, we find that

$$R_T \le \frac{B_y d \log\left(\frac{T+d}{d}\right) \left( B_y \sqrt{d \log\left(\frac{B_x T+1}{\alpha}\right)} + B_\theta \right)^2}{\epsilon^2} + |\mathcal{K}| \left( 512 B_y \log(1/\alpha) + 22 \frac{B_y \log(1/\alpha)}{L_\xi \epsilon} \right) + 36 B_y T L_\xi \epsilon.$$

Using the definition of $\mathcal{K}$, $\epsilon$ and $\alpha$ allows us to conclude the proof.

## C  Proof of Theorem 2

The proof of Theorem 2 follows closely the proof of Theorem 1. The following two Lemmas are analogues of Lemmas 1 and 2.

**Lemma 5.** *Let $\overline{\mathcal{X}}$ be an $\left(\frac{\epsilon}{3L_g}\right)^{1/\beta}$-covering of $\mathcal{B}_{B_x, d}$ of minimal cardinality, and let $\mathcal{G} = \bigcup_{\overline{x} \in \overline{\mathcal{X}}} \mathcal{G}_{\overline{x}}$.*

*Almost surely, the length of exploration phase $\mathcal{G}$ is bounded as*

$$|\mathcal{G}| \le \left( 2B_x \left( \frac{3L_g}{\epsilon} \right)^{1/\beta} + 1 \right)^d (\tau + 1).$$

Recall that we defined the event $\mathcal{E}$ as

$$\mathcal{E} = \left\{ \forall t \notin \mathcal{G}, |\widehat{g}_t - g(x_t)| \le \epsilon, \text{ and } \left| \widehat{D}_t^k - D(k\epsilon) \right| \le \sqrt{\frac{2 \log(1/\alpha)}{N_t^k}} + L_\xi \epsilon \right\}.$$

The following lemma shows that $\mathcal{E}$ happens with large probability.

**Lemma 6.** *The event $\mathcal{E}$ happens with probability at least $1 - (\alpha + 2T^2 |\mathcal{K}| \alpha)$.*

The rest of the proof holds follows the proof of Theorem 1. In particular, on the event $\mathcal{E}$, we still have

$$R_T = \sum_{t \in \mathcal{G}} \left( \max_{p \in [0, B_y]} \pi(x_t, p) - \pi(x_t, p_t) \right) + \sum_{t \notin \mathcal{G}} \left( \max_{p \in [0, B_y]} \pi(x_t, p) - \pi(x_t, p_t) \right)$$

$$\le B_y |\mathcal{G}| + |\mathcal{K}| \left( 512 B_y \log(1/\alpha) + 22 \frac{B_y \log(1/\alpha)}{L_\xi \epsilon} \right) + 36 B_y T L_\xi \epsilon.$$

where we used the fact that the instantaneous regret is bounded by $B_y$ along with Lemma 4. Using Lemma 5, we obtain

$$R_T \le B_y \left( 2B_x \left( \frac{3L_g}{\epsilon} \right)^{1/\beta} + 1 \right)^d (\tau + 1) + |\mathcal{K}| \left( 512 B_y \log(1/\alpha) + 22 \frac{B_y \log(1/\alpha)}{L_\xi \epsilon} \right) + 36 B_y T L_\xi \epsilon.$$

Using the definition of $\mathcal{K}$, $\epsilon$, $\tau$ and $\alpha$ allows us to conclude the proof.

# D    Proof of Auxilliary Lemmas

## D.1    Proof of Lemma 1

We use the elliptical potential Lemma (see, e.g., Proposition 1 in [8]) to bound the total number of rounds used to estimate $\theta$. Formally, denote the estimation indices $\mathcal{G} = \left\{ t_1 \ldots, t_{|\mathcal{G}|} \right\}$ and notice that $\iota_t = 1$ only for these indices. Thus, for all $i \in [|\mathcal{G}|]$, we can write $\mathbf{V}_{t_i} = \sum_{k=1}^{i} x_{t_k} x_{t_k}^\top + \mathbf{I}_d$ and $\mathbf{V}_{t_i - 1} = \mathbf{V}_{t_{i-1}}$. In particular, the elliptical potential lemma implies that

$$\sum_{i=1}^{|\mathcal{G}|} \|x_{t_i}\|_{\mathbf{V}_{t_i-1}^{-1}} = \sum_{i=1}^{|\mathcal{G}|} \|x_{t_i}\|_{\mathbf{V}_{t_{i-1}}^{-1}} \leq \sqrt{|\mathcal{G}| d \log \left( \frac{|\mathcal{G}| + d}{d} \right)}.$$

Since for all $t$ such that $\iota_t = 1$, $x_t^\top \mathbf{V}_{t_i-1}^{-1} x_t \geq \mu$, this implies that

$$|\mathcal{G}| \mu \leq \sqrt{|\mathcal{G}| d \log \left( \frac{|\mathcal{G}| + d}{d} \right)}.$$

Now, almost surely, $|\mathcal{G}| \leq T$. Using this bound and reorganizing the inequality leads to the desired result

$$|\mathcal{G}| \leq \frac{d \log \left( \frac{T+d}{d} \right)}{\mu^2}.$$

## D.2    Proof of Lemma 2

Lemma 2 is obtained by combining the following two results.

**Lemma 7.** *Let us define the event*

$$\mathcal{E}_1 = \{ \forall t \notin \mathcal{G} : |g(x_t) - \widehat{g}_t| \leq \epsilon \}$$

*Then, the event $\mathcal{E}_1$ happens with probability at least $1 - \alpha$.*

The remainder of the proof follows from the following lemma.

**Lemma 8.** *Let us define the event*

$$\mathcal{E} = \left\{ \forall t \in [T], k \in \mathcal{K}, \left| \widehat{D}_t^k - D(k\epsilon) \right| \leq \sqrt{\frac{2 \log(1/\alpha)}{N_t^k}} + L_\xi \epsilon \right\} \cap \mathcal{E}_1$$

*Assume that event $\mathcal{E}_1$ holds with probability $1 - \alpha$. Then, the event $\mathcal{E}$ happens with probability at least $1 - (\alpha + 2T^2 |\mathcal{K}| \alpha)$.*

## D.3    Proof of Lemma 3

We assume that $t \notin \mathcal{G}$, that $k_t = k$, and that $N_t^k > 0$ (otherwise the statement is trivial). We begin by stating an auxiliary result, which follows immediately from Lemma 2.

**Lemma 9.** *On the event $\mathcal{E}$, we have that for all $t \notin \mathcal{G}$, and all $k \in \mathcal{A}_t$;*

$$\mathrm{LCB}_t(k) \leq \pi(x_t, \widehat{g}_t + k\epsilon) \leq \mathrm{UCB}_t(k).$$

*Moreover, $k_t^* \in \mathcal{K}_t$, where*

$$k_t^* \in \arg\max_{k \in \mathcal{A}_t} \pi(x_t, \widehat{g}_t + k\epsilon).$$

On the event $\mathcal{E}$, Lemma 9 implies that

$$\pi(x_t, \widehat{g}_t + k\epsilon) \geq \mathrm{LCB}(k)$$
$$= \mathrm{UCB}(k) - (\mathrm{UCB}(k) - \mathrm{LCB}(k)).$$

Since $k_t^* \in \mathcal{A}_t$, we have

$$\mathrm{UCB}_t(k) \geq \mathrm{LCB}_t(k_t^*).$$

This implies

$$\pi(x_t, \widehat{g}_t + k\epsilon) \geq \text{LCB}_t(k_t^*) - (\text{UCB}_t(k) - \text{LCB}_t(k))$$
$$= \text{UCB}_t(k_t^*) - (\text{UCB}_t(k) - \text{LCB}_t(k)) - (\text{UCB}_t(k_t^*) - \text{LCB}_t(k_t^*))$$
$$\geq \pi(x_t, \widehat{g}_t + k_t^*\epsilon) - (\text{UCB}_t(k) - \text{LCB}_t(k)) - (\text{UCB}_t(k_t^*) - \text{LCB}_t(k_t^*))$$

Thus,

$$\pi(x_t, \widehat{g}_t + k_t^*\epsilon) - \pi(x_t, \widehat{g}_t + k\epsilon)$$
$$\leq (\text{UCB}_t(k) - \text{LCB}_t(k)) + (\text{UCB}_t(k_t^*) - \text{LCB}_t(k_t^*)).$$

Now,

$$\text{UCB}_t(k) - \text{LCB}_t(k) = (\widehat{g}_t + k\epsilon)\left(\sqrt{\frac{8\log(1/\alpha)}{N_t^k}} + 4L_\xi\epsilon\right)$$
$$\leq B_y\left(\sqrt{\frac{8\log(1/\alpha)}{N_t^k}} + 4L_\xi\epsilon\right)$$

since $k \in \mathcal{A}_t$. Moreover, since $k_t = k$, and since $k_t^* \in \mathcal{K}_t$ by Lemma 9, we know that $N_t^k \leq N_t^{k^*}$. This implies that

$$\text{UCB}_t(k_t^*) - \text{LCB}_t(k_t^*) = (\widehat{g}_t + k_t^*\epsilon)\left(\sqrt{\frac{8\log(1/\alpha)}{N_t^{k_t^*}}} + 4L_\xi\epsilon\right)$$
$$\leq B_y\left(\sqrt{\frac{8\log(1/\alpha)}{N_t^k}} + 4L_\xi\epsilon\right)$$

Thus,

$$\pi(x_t, \widehat{g}_t + k_t^*\epsilon) - \pi(x_t, \widehat{g}_t + k\epsilon) \leq 2B_y\left(\sqrt{\frac{8\log(1/\alpha)}{N_t^k}} + 4L_\xi\epsilon\right). \tag{7}$$

Next, we bound the discretization error using the following Lemma.

**Lemma 10.** *On the event $\mathcal{E}$, we have that*

$$\left|\sup_{p \in [0, B_y]} \pi(x_t, p) - \pi(x_t, \widehat{g}_t + k_t^*\epsilon)\right| \leq B_y L_\xi\epsilon.$$

By Lemma 10, Equation (7) implies that on the event $\mathcal{E}$,

$$\Delta(x_t, \widehat{g}_t + k\epsilon) \leq B_y\left(4\sqrt{\frac{2\log(1/\alpha)}{N_t^k}} + 9L_\xi\epsilon\right).$$

### D.4 Proof of Lemma 4

Note that

$$\sum_{t \notin \mathcal{G}}\left(\max_{p \in [0, B_y]} \pi(x_t, p) - \pi(x_t, p_t)\right) = \sum_{k \in \mathcal{K}} \sum_{t \notin \mathcal{G}: k_t = k} \Delta(x_t, \widehat{g}_t + k\epsilon) \tag{8}$$

We bound this term on the high-probability event $\mathcal{E}$. For $k \in \mathcal{K}$, we define $t_1^k < \cdots < t_{N_{T+1}^k}^k$ the rounds where $t \notin \mathcal{G}$ and $k_t = k$. We split these rounds into episodes as follows. We define $\bar{a} = \lfloor -\log_2(18L_\xi\epsilon)\rfloor$. For $a \in [\![1, \bar{a}]\!]$, we also define

$$t_a = \frac{128\log(1/\alpha)}{2^{-2a}}.$$

With these notations, we have

$$\sum_{t\notin\mathcal{G}:k_t=k}\Delta(x_t,\widehat{g}_t+k\epsilon)=\sum_{i\leq \mathfrak{t}_1\wedge N^k_{T+1}}\Delta(x_{t^k_i},\widehat{g}_{t^k_i}+k\epsilon)+\sum_{a=1}^{\overline{a}-1}\sum_{\mathfrak{t}_a\wedge N^k_{T+1}<i\leq \mathfrak{t}_{a+1}\wedge N^k_{T+1}}\Delta(x_{t^k_i},\widehat{g}_{t^k_i}+k\epsilon)$$

$$+\sum_{\mathfrak{t}_{\overline{a}}\wedge N^k_{T+1}<i\leq N^k_{T+1}}\Delta(x_{t^k_i},\widehat{g}_{t^k_i}+k\epsilon)$$

On the one hand, $\Delta(x_t,p_t)\leq B_y$ for all $t\leq T$, so

$$\sum_{i\leq \mathfrak{t}_1\wedge N^k_{T+1}}\Delta(x_{t^k_i},\widehat{g}_{t^k_i}+k\epsilon)\leq B_y\mathfrak{t}_1$$

On the other hand, using Lemma 3, we see that on the event $\mathcal{E}$, if $i\geq \mathfrak{t}_a$ and $a\in[\![1,\overline{a}]\!]$,

$$\Delta(x_{t^k_i},\widehat{g}_{t^k_i}+k\epsilon)\leq B_y\left(4\sqrt{\frac{2\log(1/\alpha)}{\mathfrak{t}_a}}+9L_\xi\epsilon\right)$$

$$\leq B_y\left(\frac{2^{-a}}{2}+9L_\xi\epsilon\right)$$

Since $2^{-a}\geq 18L_\xi\epsilon$, this implies that

$$\Delta(x_{t^k_i},\widehat{g}_{t^k_i}+k\epsilon)\leq 2^{-a}B_y.$$

Then,

$$\sum_{a=1}^{\overline{a}-1}\sum_{\mathfrak{t}_a\wedge N^k_{T+1}<i\leq \mathfrak{t}_{a+1}\wedge N^k_{T+1}}\Delta(x_{t^k_i},\widehat{g}_{t^k_i}+k\epsilon)\leq B_y\sum_{a=1}^{\overline{a}-1}\left(\mathfrak{t}_{a+1}-\lceil \mathfrak{t}_a\rceil+1\right)2^{-a}$$

$$\leq B_y\sum_{a=1}^{\overline{a}-1}\left(\mathfrak{t}_{a+1}-\mathfrak{t}_a\right)2^{-a}+B_y$$

By definition of $\mathfrak{t}_a$, this implies that

$$\sum_{a=1}^{\overline{a}-1}\sum_{\mathfrak{t}_a\wedge N^k_{T+1}<i\leq \mathfrak{t}_{a+1}\wedge N^k_{T+1}}\Delta(x_{t^k_i},\widehat{g}_{t^k_i}+k\epsilon)\leq 128B_y\log(1/\alpha)\sum_{a=1}^{\overline{a}-1}\left(2^{2a+2}-2^{2a}\right)2^{-a}+B_y$$

$$\leq 384B_y\log(1/\alpha)\left(1+\sum_{a=1}^{\overline{a}-1}2^a\right)$$

$$\leq 384B_y\log(1/\alpha)2^{\overline{a}}$$

$$\leq 22\frac{B_y\log(1/\alpha)}{L_\xi\epsilon}$$

where we used that $2^{\overline{a}}\leq\frac{1}{18L_\xi\epsilon}$. Similarly,

$$\sum_{\mathfrak{t}_{\overline{a}}\wedge N^k_{T+1}<i\leq N^k_{T+1}}\Delta(x_{t^k_i},\widehat{g}_{t^k_i}+k\epsilon)\leq 2^{-\overline{a}}B_yN^k_{T+1}$$

$$\leq 36B_yN^k_{T+1}L_\xi\epsilon.$$

Combining these results, we find that

$$\sum_{t\notin\mathcal{G}:k_t=k}\Delta(x_t,\widehat{g}_t+k\epsilon)\leq 512B_y\log(1/\alpha)+22\frac{B_y\log(1/\alpha)}{L_\xi\epsilon}+36B_yN^k_{T+1}L_\xi\epsilon.\qquad(9)$$

We conclude the proof by summing over $k\in\mathcal{K}$, and using the fact that $\sum_{k\in\mathcal{K}}N^k_{T+1}\leq T$.

## D.5 Proof of Lemma 5

We note that

$$|\mathcal{G}| \le |\overline{\mathcal{X}}|(\tau + 1).$$

We conclude by using classical results on covering number of the ball (see, e.g., Corollary 4.2.13 in [29]), stating that there exists an $\left(\frac{\epsilon}{3L_g}\right)^{1/\beta}$-covering of the ball of radius $B_x$ in dimension $d$ of cardinality at most $\left(2B_x \left(\frac{3L_g}{\epsilon}\right)^{1/\beta} + 1\right)^d$.

## D.6 Proof of Lemma 6

The proof of Lemma 6 relies on the following Lemma.

**Lemma 11.** *Let us define the event*

$$\mathcal{E}_1 = \{\forall t \notin \mathcal{G} : |g(x_t) - \widehat{g}(\overline{x}_t)| \le \epsilon\}$$

*Then, the event $\mathcal{E}_1$ happens with probability at least $1 - \alpha$.*

Note that Lemma 8 still holds for non-parametric valuations. This concludes the proof of Lemma 6.

## D.7 Proof of Lemma 7

We introduce the variables

$$\tilde{x}_t = \iota_t x_t \quad \text{and} \quad \tilde{y}_t = 2B_y \iota_t \left(o_t - \frac{1}{2}\right)$$

and the $\sigma$-algebra $\mathcal{F}_t = \sigma\left((x_s)_{s \le t+1}, (o_s)_{s \le t}\right)$. Since $\mathbf{V}_{t-1}$ and $x_t$ are $\mathcal{F}_{t-1}$-measurable, then so does $\iota_t$, and thus both $\tilde{x}_{t+1}$ and $\tilde{y}_t$ are $\mathcal{F}_t$-measurable. Moreover, for any round where $\iota_t = 1$, the price is chosen uniformly at random and we have

$$
\begin{aligned}
\mathbb{E}\left[\tilde{y}_t | \mathcal{F}_{t-1}\right] &= \iota_t \times \left(2B_y \int_{-B_y}^{B_y} \mathbb{P}\left[u \le y_t | \mathcal{F}_{t-1}\right] \frac{\mathrm{d}u}{2B_y} - B_y\right) \\
&= \iota_t \times \left(\int_{-B_y}^{B_y} \int_{-B_\xi}^{B_\xi} \mathbb{1}\left\{u \le x_t^\top \theta + \xi\right\} f(\xi) \, \mathrm{d}\xi \, \mathrm{d}u - B_y\right) \\
&= \iota_t \times \left(\int_{-B_\xi}^{B_\xi} \int_{-B_y}^{\xi + x_t^\top \theta} \mathrm{d}u f(\xi) \, \mathrm{d}\xi - B_y\right) \\
&= \iota_t \times \left(x_t^\top \theta + \int_{-B_\xi}^{B_\xi} \xi f(\xi) \, \mathrm{d}\xi\right) \\
&= \iota_t \times x_t^\top \theta
\end{aligned}
$$

where in the last equality we used that $\int_{-B_\xi}^{B_\xi} \xi f(\xi) \, \mathrm{d}\xi = \mathbb{E}\left[\xi_t\right] = 0$. The same relation also trivially holds when $\iota_t = 0$. Thus, conditionally on $\mathcal{F}_{t-1}$, $\tilde{y}_t - \tilde{x}_t^\top \theta$ is centered and in $[-B_y, B_y]$, which implies that it is $B_y$-subgaussian. Now, for all $t \le T$, we have

$$
\begin{aligned}
\widehat{\theta}_t &= 2B_y \left(\sum_{s<t} \iota_s x_s x_s^\top + \mathbf{I}_d\right)^{-1} \sum_{s \in \mathcal{G}} \left(o_s - \frac{1}{2}\right) x_s \\
&= \left(\sum_{s<t} \tilde{x}_s \tilde{x}_s^\top + \mathbf{I}_d\right)^{-1} \sum_{s<t} \tilde{y}_s \tilde{x}_s.
\end{aligned}
$$

Using the fact that for all $t \ge 1$, $\|\tilde{x}_t\| \le B_x$, and that $\|\theta\| \le B_\theta$, and applying Theorem 2 in [1], we find that for all $t \ge 0$, with probability $1 - \alpha$,

$$\|\widehat{\theta}_t - \theta\|_{(\sum_{s<t} \tilde{x}_l \tilde{x}_l^\top + \mathbf{I}_d)} \le B_y \sqrt{d \log\left(\frac{1 + B_x^2 T}{\alpha}\right)} + B_\theta.$$

Note that our definitions of $\tilde{x}_t$ and $\tilde{y}_t$ ensure that $\|\widehat{\theta}_t - \theta\|_{(\sum_{s<t} \tilde{x}_l \tilde{x}_l^\top + \mathbf{I}_d)} = \|\widehat{\theta}_t - \theta\|_{\mathbf{V}_t}$. Moreover, for all $t$,

$$|x_t^\top (\widehat{\theta}_t - \theta)| \le \|x_t^\top\|_{\mathbf{V}_t^{-1}} \|\widehat{\theta}_t - \theta\|_{\mathbf{V}_t}.$$

In particular, if $t \notin \mathcal{G}$, $\|x_t^\top\|_{(\mathbf{V}_t)^{-1}} \le \mu$, so

$$|x_t^\top (\widehat{\theta}_t - \theta)| \le \mu \left( B_y \sqrt{d \log\left(\frac{1 + B_x^2 T}{\alpha}\right)} + B_\theta \right).$$

The conclusion follows from the choice $\epsilon = \mu \left( B_y \sqrt{d \log\left(\frac{1 + B_x^2 T}{\alpha}\right)} + B_\theta \right)$, and the fact that $\widehat{g}_t = x_t^\top \widehat{\theta}_t$.

### D.8 Proof of Lemma 8

We rely on the following well-known result (we provide proof in the appendix for the sake of completeness).

**Lemma 12.** *Let $(y_t)_{t \ge 1}$ be a sequence of random variables adapted for a filtration $\mathcal{F}_t$, such that $y_t - \mathbb{E}[y_t | \mathcal{F}_{t-1}] \in [m, M]$. Assume that for $t \in \mathbb{N}_*$, $\iota_t \in \{0, 1\}$ is $\mathcal{F}_{t-1}$-measurable, and define $N_t = \sum_{s \le t} \iota_s$, and $\widehat{\mu}_t = \frac{\sum_{s \le t} \iota_s (y_s - \mathbb{E}[y_s | \mathcal{F}_{s-1}])}{N_t}$ if $N_t \ge 1$. Then, for any $t \in \mathbb{N}_*$ and $\alpha \in (0, 1)$,*

$$\mathbb{P}\left( N_t = 0 \text{ or } |\widehat{\mu}_t| \le (M - m) \sqrt{\frac{\log(1/\alpha)}{2 N_t}} \right) \ge 1 - 2t\alpha.$$

*Moreover, for any $l > 0$ and $\alpha \in (0, 1)$,*

$$\mathbb{P}\left( N_t = l \text{ and } |\widehat{\mu}_t| \ge (M - m) \sqrt{\frac{\log(1/\alpha)}{2 N_t}} \right) \le 2\alpha.$$

Note Lemma 8 holds trivially for all $t$ such that $N_t^k = 0$. Therefore we assume w.l.o.g. that $N_t^k \ge 1$ (otherwise the statement is trivial). For any such given $t \in [T]$, we control the error $|\widehat{F}_t^k - F(k\epsilon)|$ uniformly for $k \in \mathcal{K}$. To do so, we rely on Lemma 12; we define $\tilde{\iota}_t = \mathbb{1}\{\iota_t = 0 \text{ and } k_t = k\}$, and note that for $\mathcal{F}_t = \sigma((x_1, \ldots, x_{t+1}), (o_1, \ldots, o_t))$, $\tilde{\iota}_t$ is $\mathcal{F}_{t-1}$-measurable, and $o_t$ is $\mathcal{F}_t$ adapted. Moreover,

$$\tilde{\iota}_t \mathbb{E}[o_t | \mathcal{F}_{t-1}] = \tilde{\iota}_t \mathbb{P}(g(x_t) + \xi_t \ge \widehat{g}_t + k\epsilon)$$
$$= \tilde{\iota}_t D(\widehat{g}_t - g(x_t) + k\epsilon),$$

and directly by definition, it holds that $\widehat{D}_t^k = \frac{\sum_{s \le t} \tilde{\iota}_t o_t}{N_t}$. Using Lemma 12, we find that with probability $1 - 2\alpha t$, $N_t^k = 0$ or

$$\left| \widehat{D}_t^k - \frac{\sum_{s \le t} \tilde{\iota}_t D(\widehat{g}_t - g(x_t) + k\epsilon)}{N_t^k} \right| \le \sqrt{\frac{2 \log(1/\alpha)}{N_t^k}}.$$

Moreover, on the event $\mathcal{E}_1$, which happens w.p. at least $1 - \alpha$, for all $t \notin \mathcal{G}$, $|\widehat{g}_t - g(x_t)| \le \epsilon$. Using the fact that $D$ is $L_\xi$-Lipschitz, we find that for all $t \notin \mathcal{G}$,

$$|D(\widehat{g}_t - g(x_t) + k\epsilon) - D(k\epsilon)| \le L_\xi |\widehat{g}_t - g(x_t)| \le L_\xi \epsilon.$$

Thus, with probability $1 - 2\alpha t$,

$$\left| \widehat{D}_t^k - D(k\epsilon) \right| \le \sqrt{\frac{2 \log(1/\alpha)}{N_t^k}} + L_\xi \epsilon.$$

Using a union bound over all $k \in \mathcal{K}$ and $t \in [T]$ and then intersecting with $\mathcal{E}_1$ using another union bound yields the desired result.

## D.9 Proof of Lemma 9

For any $t \notin \mathcal{G}$, denoting $p_t(k) = \widehat{g}_t + k\epsilon$, we first rewrite

$$
\begin{aligned}
\pi(x_t, p_t(k)) &= \mathbb{E}[p_t(k)\mathbb{1}\{p_t(k) \le y_t\}|p_t(k), x_t] \\
&= p_t(k)\mathbb{E}[\mathbb{1}\{p_t(k) \le g(x_t) + \xi_t\}|p_t(k), x_t] \\
&= p_t(k)D(p_t(k) - g(x_t)) \\
&= (\widehat{g}_t + k\epsilon)D(\widehat{g}_t - g(x_t) + k\epsilon) \\
&= (\widehat{g}_t + k\epsilon)\widehat{D}_t^k + (\widehat{g}_t + k\epsilon)\Big(D(\widehat{g}_t - g(x_t) + k\epsilon) - \widehat{D}_t^k\Big).
\end{aligned}
$$

Since the event $\mathcal{E}$ holds, the following hold for all $t \notin \mathcal{G}$ and $k \in \mathcal{A}_t$:

$$
|\widehat{g}_t - g(x_t)| \le \epsilon, \qquad \text{and} \qquad \left|\widehat{D}_t^k - D(k\epsilon)\right| \le \sqrt{\frac{2\log(1/\alpha)}{N_t^k}} + L_\xi\epsilon.
$$

In particular, we have that:

$$
\begin{aligned}
\left|D(\widehat{g}_t - g(x_t) + k\epsilon) - \widehat{D}_t^k\right| &\le |D(\widehat{g}_t - g(x_t) + k\epsilon) - D(k\epsilon)| + \left|D(k\epsilon) - \widehat{D}_t^k\right| \\
&\stackrel{(1)}{\le} L_\xi|\widehat{g}_t - g(x_t)| + \left|D(k\epsilon) - \widehat{D}_t^k\right| \\
&\le L_\xi\epsilon + \sqrt{\frac{2\log(1/\alpha)}{N_t^k}} + L_\xi\epsilon \\
&= \sqrt{\frac{2\log(1/\alpha)}{N_t^k}} + 2L_\xi\epsilon
\end{aligned}
$$

Relation (1) holds since $D$ is $L_\xi$-Lipschitz and (2) is under the event $\mathcal{E}$ for all $t \notin \mathcal{E}$. As the set $\mathcal{A}_t$ is chosen such that $\widehat{g}_t + k\epsilon \ge 0$ for all $k \in \mathcal{A}_t$, it implies that

$$
\left|\pi(x_t, \widehat{g}_t + k\epsilon) - (\widehat{g}_t + k\epsilon)\widehat{D}_t^k\right| \le (\widehat{g}_t + k\epsilon)\left(\sqrt{\frac{2\log(1/\alpha)}{N_t^k}} + 2L_\xi\epsilon\right).
$$

Reorganizing, we get for all $k \in \mathcal{A}_t$ and $t \notin \mathcal{G}$

$$
\text{LCB}_t(k) \le \pi(x_t, \widehat{g}_t + k\epsilon) \le \text{UCB}_t(k).
$$

which proves the first part of the statement.

Now let $k_t^* \in \arg\max_{k \in \mathcal{A}_t} \pi(x_t, \widehat{g}_t + k\epsilon)$. By the first part of the claim, it holds that

$$
\text{UCB}_t(k_t^*) \stackrel{(*)}{\ge} \pi(x_t, \widehat{g}_t + k_t^*\epsilon) = \max_{k \in \mathcal{A}_t} \pi(x_t, \widehat{g}_t + k\epsilon) \stackrel{(*)}{\ge} \max_{k \in \mathcal{A}_t} \text{LCB}_t(k),
$$

where relations $(*)$ are due to the first part of the lemma; this proves that $k_t^* \in \mathcal{K}_t$.

## D.10 Proof of Lemma 10

The proof follows by noticing that, on the one hand, $\mathcal{K}$ ensures that for all $p \in [0, B_y]$, there exists $k \in \mathcal{K}$ such that $\widehat{g}_t + k\epsilon \in [0, B_y]$ and $|\widehat{g}_t + k\epsilon - p| \le \epsilon$. On the other hand, the prices considered are bounded by $B_y$, and the demand function $D$ is $L_\xi$-Lipschitz, so the reward function $\pi$ is $B_y L_\xi$-Lipschitz.

## D.11 Proof of Lemma 11

For $\overline{x} \in \mathcal{X}$, let us define recursively the variables $\iota_1^{\overline{x}} = \mathbb{1}\{\overline{x}_1 = \overline{x}\}$, and for $t > 1$, $\iota_t^{\overline{x}} = \mathbb{1}\{\overline{x}_t = \overline{x}, \text{ and } \sum_{s<t} \iota_s^{\overline{x}} < \tau\}$, and define the variables

$$
\widetilde{g}_t^{\overline{x}} = \iota_t^{\overline{x}} g(x_t) \quad \text{and} \quad \widetilde{y}_t^{\overline{x}} = 2B_y \iota_t^{\overline{x}}\left(o_t - \frac{1}{2}\right)
$$

and the $\sigma$-algebra $\mathcal{F}_t = \sigma\left((x_s)_{s \leq t+1}, (o_s)_{s \leq t}\right)$. Note that $\iota_t^{\overline{x}}$ is $\mathcal{F}_{t-1}$-measurable, and thus both $\tilde{x}_{t+1}$ and $\tilde{y}_t$ are $\mathcal{F}_t$-measurable. Moreover, for any round where $\iota_t^{\overline{x}} = 1$, the price is chosen uniformly at random and we have

$$
\begin{aligned}
\mathbb{E}\left[\tilde{y}_t^{\overline{x}} | \mathcal{F}_{t-1}\right] &= \iota_t^{\overline{x}} \times \left(2B_y \int_{-B_y}^{B_y} \mathbb{P}\left[u \leq y_t | \mathcal{F}_{t-1}\right] \frac{\mathrm{d}u}{2B_y} - B_y\right) \\
&= \iota_t^{\overline{x}} \times \left(\int_{-B_y}^{B_y} \int_{-B_\xi}^{B_\xi} \mathbb{1}\left\{u \leq g(x_t) + \xi\right\} f(\xi) \, \mathrm{d}\xi \, \mathrm{d}u - B_y\right) \\
&= \iota_t^{\overline{x}} \times \left(\int_{-B_\xi}^{B_\xi} \int_{-B_y}^{\xi + g(x_t)} \mathrm{d}u f(\xi) \, \mathrm{d}\xi - B_y\right) \\
&= \iota_t^{\overline{x}} \times \left(g(x_t) + \int_{-B_\xi}^{B_\xi} \xi f(\xi) \, \mathrm{d}\xi\right) \\
&= \tilde{g}_t^{\overline{x}}
\end{aligned}
$$

where in the last equality we used that $\int_{-B_\xi}^{B_\xi} \xi f(\xi) \, \mathrm{d}\xi = \mathbb{E}\left[\xi_t\right] = 0$. The same relation also trivially holds when $\iota_t^{\overline{x}} = 0$. Thus, conditionally on $\mathcal{F}_{t-1}$, $\tilde{y}_t - \tilde{g}_t^{\overline{x}}$ is centered and in $[-B_y, B_y]$. We denote $N_t^{\overline{x}} = \sum_{s < t} \iota_s^{\overline{x}}$, we note that if $t \notin \mathcal{G}^{\overline{x}}$, then $N_t^{\overline{x}} = \lceil \tau \rceil$ a.s. Using Lemma 12, we find that for all $t \notin \mathcal{G}^{\overline{x}}$, a.s., $N_t^{\overline{x}} = \lceil \tau \rceil$. Then,

$$
\begin{aligned}
\mathbb{P}\left(\exists t \notin \mathcal{G}^{\overline{x}} : \left|\frac{\sum_{s \in \mathcal{G}^{\overline{x}}, s < t} \tilde{y}_t^{\overline{x}} - \tilde{g}_t^{\overline{x}}}{N_t^{\overline{x}}}\right| \geq 2B_y \sqrt{\frac{\log(2|\overline{\mathcal{X}}|/\alpha)}{2\lceil\tau\rceil}}\right) & \\
\leq \mathbb{P}\left(N_t^{\overline{x}} = \lceil\tau\rceil \text{ and } \left|\frac{\sum_{s \in \mathcal{G}^{\overline{x}}, s < t} \tilde{y}_t^{\overline{x}} - \tilde{g}_t^{\overline{x}}}{N_t^{\overline{x}}}\right| \geq 2B_y \sqrt{\frac{\log(2|\overline{\mathcal{X}}|/\alpha)}{2\lceil\tau\rceil}}\right) & \\
\leq \frac{\alpha}{|\overline{\mathcal{X}}|}. &
\end{aligned}
$$

Moreover, since $g$ is $(L_g, \beta)$-Hölder- continuous, and $\|\overline{x}_t - x_t\| \leq \left(\frac{\epsilon}{3L_g}\right)^{1/\beta}$ a.s., we have

$$
|g(\overline{x}_t) - \tilde{g}_t^{\overline{x}}| \leq L_g \cdot \left[\left(\frac{\epsilon}{3L_g}\right)^{1/\beta}\right]^\beta = \frac{\epsilon}{3}.
$$

Then, with probability at least $1 - \alpha/|\overline{\mathcal{X}}|$, for all $t \notin \mathcal{G}^{\overline{x}}$,

$$
\begin{aligned}
|\widehat{g}(\overline{x}_t) - g(\overline{x}_t)| &\leq 2B_y \sqrt{\frac{\log(2|\overline{\mathcal{X}}|/\alpha)}{2\lceil\tau\rceil}} + \frac{\epsilon}{3} \\
&\leq \frac{2\epsilon}{3}.
\end{aligned}
$$

where we used $\tau = \frac{18B_y^2 \log(|\overline{\mathcal{X}}|/\alpha)}{\epsilon^2}$. Using a union bound over $\overline{\mathcal{X}}$, we find that with probability at least $1 - \alpha$, for all $t \notin \mathcal{G}^{\overline{x}}$,

$$
|\widehat{g}(\overline{x}_t) - g(\overline{x}_t)| \leq \frac{2\epsilon}{3}.
$$

Similarly, for all $t \notin \mathcal{G}$, $\|g(x_t) - g(\overline{x}_t)\| \leq L_g \frac{\epsilon}{3L_g}$. Then, we have that with probability $1 - \alpha$, for all $t \notin \mathcal{G}^{\overline{x}}$,

$$
|\widehat{g}(\overline{x}_t) - g(x_t)| \leq \epsilon.
$$

## D.12 Proof of Lemma 12

Let us define $Z_t = \sum_{s \leq t} \iota_s(y_s - \mathbb{E}[y_s|\mathcal{F}_{s-1}])$, and for $x \in \mathbb{R}$, $M_t = \exp\left(xZ_t - \frac{x^2(M-m)^2 N_t}{8}\right)$.
We begin by showing that $M_t$ is a super-martingale. Indeed, we have that

$$\mathbb{E}\left[e^{x\iota_t(y_t - \mathbb{E}[y_t|\mathcal{F}_{t-1}])}\Big|\mathcal{F}_{t-1}\right] = \mathbb{E}\left[\iota_t e^{x(y_t - \mathbb{E}[y_t|\mathcal{F}_{t-1}])} + (1-\iota_t)\Big|\mathcal{F}_{t-1}\right]$$

$$\leq \iota_t e^{\frac{x^2(M-m)^2}{8}} + (1-\iota_t)$$

$$\leq e^{\frac{x^2(M-m)^2 \iota_t}{8}}.$$

where we use the fact that $(y_t - \mathbb{E}[y_t|\mathcal{F}_{t-1}])$ is bounded in $[m, M]$ together with the conditional version of Hoeffding's Lemma. Noticing that

$$M_t = M_{t-1} e^{x\iota_t(y_t - \mathbb{E}[y_t|\mathcal{F}_{t-1}]) - \frac{x^2(M-m)^2 \iota_t}{8}},$$

this proves that $M_t$ is a super-martingale, and so $\mathbb{E}[M_t] \leq \mathbb{E}[M_0] = 1$.

Now, for all $\epsilon > 0$ and all $l \in \mathbb{N}$, and all $x > 0$, by a Markov-Chernoff argument,

$$\mathbb{P}(Z_t \geq \epsilon \text{ and } N_t = l) = \mathbb{P}\left(\mathbb{1}\{N_t = l\}e^{xZ_t} \geq e^{\epsilon x}\right)$$

$$\leq e^{-\epsilon x}\mathbb{E}\left(e^{xZ_t}\mathbb{1}\{N_t = l\}\right)$$

$$= e^{-\epsilon x + \frac{x^2(M-m)^2 l}{8}}\mathbb{E}\left(e^{xZ_t - \frac{x^2(M-m)^2 l}{8}}\mathbb{1}\{N_t = l\}\right).$$

Using the previous result, we have that

$$\mathbb{E}\left(e^{xZ_t - \frac{x^2(M-m)^2 l}{8}}\mathbb{1}\{N_t = l\}\right) = \mathbb{E}\left(e^{xZ_t - \frac{x^2(M-m)^2 N_t}{8}}\mathbb{1}\{N_t = l\}\right)$$

$$\leq \mathbb{E}\left(e^{xZ_t - \frac{x^2(M-m)^2 N_t}{8}}\right)$$

$$= \mathbb{E}(M_t)$$

$$\leq \mathbb{E}(M_0) = 1.$$

so

$$\mathbb{P}(Z_t \geq \epsilon \text{ and } N_t = l) \leq e^{-\epsilon x + \frac{x^2(M-m)^2 l}{8}}.$$

In particular, for $\epsilon = (M-m)\sqrt{\frac{l \cdot \log(1/\alpha)}{2}}$ and $x = \frac{4\epsilon}{l(M-m)^2}$,

$$\mathbb{P}\left(Z_t \geq (M-m)\sqrt{\frac{l \cdot \log(1/\alpha)}{2}} \text{ and } N_t = l\right) \leq \alpha.$$

This proves the first part of the Lemma. Summing over the values of $l$ from $1$ to $t$, we find that

$$\mathbb{P}\left(Z_t \geq (M-m)\sqrt{\frac{N_t \log(1/\alpha)}{2}} \text{ and } N_t \geq 1\right) \leq t\alpha.$$

Similar arguments can be used to prove that

$$\mathbb{P}\left(-Z_t \geq (M-m)\sqrt{\frac{N_t \log(1/\alpha)}{2}} \text{ and } N_t \geq 1\right) \leq t\alpha.$$

Noting that $Z_t = \hat{\mu}_t N_t$ and normalizing by $N_t$ (and since adding the case $N_t = 0$ can only increase the probability) concludes the proof of the Lemma.

