# OpenReview forum: "Improved Algorithms for Contextual Dynamic Pricing"
_NeurIPS.cc/2024/Conference — NeurIPS 2024 poster_

### Official Review · Reviewer_iWzp · 2024-06-20

**Soundness:** 3
**Presentation:** 3
**Contribution:** 3
**Rating:** 6
**Confidence:** 3

**Summary:**

This paper expands the line of work on contextual dynamic pricing where the buyer's valuation may be noisy. The authors give a simple way to obtain an unbiased estimate for the true valuation function, and propose an exploration - exploitation type algorithm that learn simultaneously the true valuation function and the noise distribution. Based on this observation, algorithms are proposed for when the valuation function is a linear model or when it is Holder continuous.

**Strengths:**

The paper is well-written and I found it easy to follow. The necessary intuitions are adequately given. The main idea behind the algorithm is conceptually simple.

The results in this work expands the line of work on contextual dynamic pricing, though I did not check the proof, I believe the results are new and correct.

**Weaknesses:**

I think for me, the most limiting assumption is that the noise is i.i.d. across all contexts, which the authors also mentioned.

It may not be possible to add experiments now, but I think it will be helpful for this paper.

**Questions:**

In assumption 2, it is assumed the noise is bounded. Do all prior work assume that the noise is bounded? Is there any work that does not?

This is minor, but I am not sure what the green / red color signify in table 1. Also you made the assumption that the noise is bounded, but this did not seem reflected in the rows where you presented your results. The table also has margin issues.

**Limitations:**

Yes.

---

> ### Author Rebuttal · Authors · 2024-08-06
>
> We thank the reviewer for the comment.
>
> ### Weaknesses
>
> **i.i.d noise:** This assumption is standard in the literature investigating the problem of dynamic pricing. For instance, all the papers mentioned in Table 1 share such noise structure. Nevertheless, as we mentioned in the conclusion, it could be of great interest to explore other assumptions, including a framework in which it depends on the context observed. The main reason that this assumption is needed is that the distribution itself determines the optimal price (and not only, say, its median or its mean), and therefore it has to be learned. If the distribution changes between contexts, all context-dependent distributions need to be learned, making the problem significantly more difficult in terms of regret.
>
> **Experiments:** We will add experiments to the final version of the paper (as we elaborate in the response to reviewer EQLG).
>
> ### Questions
>
> **Bounded noise:** Indeed, having bounded noise is standard in the dynamic-pricing literature. Usually it is well-motivated by real-life applications - it is reasonable to consider that valuations for certain goods are bounded, which implies boundedness of the noise. Again referring to Table 1, all works make this assumption, with the exception of Cohen et al. 2019. However, their results are obtained either under the assumption that the noise is subgaussian with a very small variance factor (typically, $1/T$, so that the noise is supported on a very small interval with large probability, and the valuations are “almost deterministic”), or that the distribution of the noise is known. Both these assumptions are stronger than ours. In other works (eg. [14], [15], [21], [30]), as in ours, boundedness of the noise is necessary to ensure that the buyers’ evaluations live in a bounded interval.
>
> **Table:** The colors in the table on the left column are meant to highlight the assumption on the noise distribution and/or the reward function that are either more restrictive, marked in red, or match, in green, the one required for our algorithm. In the right column, our rates are in green, while worse rates obtained under the same assumptions are in red. We will make sure to add a sentence in the caption of the table to make this clear. We will also address the margin issues, and we thank the reviewer for bringing them to our attention.

---

### Official Review · Reviewer_B94q · 2024-07-08

**Soundness:** 3
**Presentation:** 3
**Contribution:** 3
**Rating:** 7
**Confidence:** 3

**Summary:**

The paper studies the contextual dynamic pricing problem with binary demands and an unknown noise distribution. It presents a general framework for a pricing algorithm and proves its regret bounds for two specific types of valuation functions: linear and non-parametric. In the linear case, its results improve existing results and match the lower bound.

**Strengths:**

The paper is well-written and introduces a novel framework for solving and analyzing the contextual dynamic pricing problem with unknown noise distribution. This new framework can be applied in various settings and leads to algorithms with improved regret upper bounds in the linear case.

**Weaknesses:**

One potential concern is the paper's similarity with [1] in the exploration part and its contribution in this part. Line 186 mentions, “this method (using random pricing to construct unbiased estimates of $g(x_t)$) appears to have never been used in dynamic pricing” but as mentioned in line 195, if I understand correctly, [1] (where $g(x_t)$ is a linear function) also uses random pricing to construct such unbiased estimates. Although this paper’s main algorithm and analyses are different from [1]’s, a detailed discussion on the difference in using random prices as an exploration policy would be helpful to better understand its contribution.

Another minor weakness is the lack of experimental results. Numerical experiments could help validate the efficiency of the algorithms and demonstrate their potential advantages over existing methods.

[1]Fan, Jianqing, Yongyi Guo, and Mengxin Yu. "Policy optimization using semiparametric models for dynamic pricing." Journal of the American Statistical Association119.545 (2024): 552-564.

**Questions:**

Please check point (1) in weaknesses. Additionally, compared to [1], why do the proposed algorithms achieve better regret bounds with milder assumptions (e.g., Assumption 1 does not require i.i.d. contexts as in [1])? Is this improvement due to using bandit techniques (e.g., price/action eliminations) or other reasons? More explanation would be appreciated.

**Limitations:**

See weaknesses.

---

> ### Author Rebuttal · Authors · 2024-08-06
>
> We thank the reviewer for the feedback.
>
> ### Weaknesses:
>
> Regarding the statement made in line 186  and the differences with the work in [1] we kindly refer the reviewer to the second question in the response to reviewer Beqc, where we provide more explanation regarding what we meant by this statement and we propose to remove it, having acknowledged how it might appear misleading. Nonetheless, the two methods are not identical and, in particular, ours requires fewer assumptions as we will further explain below (for a more detailed comparison of the assumptions between our papers and [1], please refer to the second question in the response to reviewer EQLG).
>
> Experiments: we will add experiments to the final version of the paper (as we elaborate in the response to reviewer EQLG).
>
> ### Question:
>
> Although our algorithm and the one proposed in [1] look similar, they present some fundamental differences that cannot be reconducted to mere technical details.
> What they propose is, in fact, an explore-exploit kind of strategy where in the first phase they gather information, by posting prices uniformly at random to build an estimate of the parameter $\theta$ and the c.d.f. $F$, which are later on used to find the optimal price. Since their exploration phase is concentrated all at once in the initial part of their algorithm, they need to make sure to receive diverse context samples to estimate the parameter $\theta$ with good precision, hence the reason for their more limiting assumption on the i.i.d. nature of the contexts, as well as the lower bound on the eigenvalues of the covariance matrix.
>
> Our strategy instead works adaptively, alternating between the value-approximation subroutine and the price-elimination one depending on the contexts received. On one side, this allows us to handle adversarial contexts, making our result more general; on the other, this minimizes the time spent in the exploration phase, and consequently, the regret accumulated.
> As mentioned initially, the second phase of the algorithm in [1] consists of an exploitation of the information gathered before, which simply posts the optimal price with respect to the approximation of the expected reward function. To estimate this reward function, the authors estimate as an intermediate step the virtual value function. By contrast, our algorithm bypasses this intermediate step altogether, making it simpler.
> To ensure convergence of the greedy price to the best price, the authors rely on assumption 2.1, which states the unimodality of the expected reward function. This assumption strongly limits the class of distributions for which this result is applicable. Without the unimodality assumption, their explore-exploit approach leads to $T^{3/4}$ regret, as discussed in Section 2.3 of our paper. In our price-elimination phase, we apply a multi-armed-bandit algorithm, which performs an extensive exploration across the possible price increments, mitigating the harmful effects that such exploration might have on the regret by allowing for the sharing of information among different contexts. This allows us to recover optimal regret rates while removing the unimodality assumption.
>
> ----
>
> [1] Fan, Jianqing, Yongyi Guo, and Mengxin Yu. "Policy optimization using semiparametric models for dynamic pricing." JASA, 2024

---

> > ### Comment · Reviewer_B94q · 2024-08-11
> >
> > Thank you for your response. It addresses my concerns and I have raised the score.

---

### Official Review · Reviewer_EQLG · 2024-07-10

**Soundness:** 3
**Presentation:** 3
**Contribution:** 3
**Rating:** 6
**Confidence:** 3

**Summary:**

The paper addresses the problem of dynamic pricing using contextual information, aiming to maximize a seller's revenue by setting prices based on the context and buyer valuations. Buyers purchase products if the prices are lower than their valuations. The authors propose an algorithm called VALUATION APPROXIMATION - PRICE ELIMINATION (VAPE) and analyze its performance under two valuation models: the linear valuation model and the non-parametric valuation model.

**Strengths:**

- The paper is well-written and easy to follow.

- The work is well-motivated from the practical perspective.

- The theoretical results seem to be correct, even if I did not check the proofs carefully.

- The related works are properly discussed.

**Weaknesses:**

- A paper with such practical motivation would benefit greatly from being accompanied by a thorough experimental campaign.

- See questions below.

**Questions:**

- In the definition of regret (between lines 96 and 97), it seems that the optimal price (i.e., the one in the max) is unique for every product. I expect that every context presents an optimal price. Do the authors agree?

- Lines 277-278: can the authors provide some more comments on the results and compare the assumptions of the other works to the one in this paper?

**Limitations:**

No limitations.

---

> ### Author Rebuttal · Authors · 2024-08-06
>
> We thank the reviewer for the comment.
>
> ### Weaknesses
>
> **Experiments:** We agree with the reviewer that it would be beneficial to further demonstrate the advantages of our approach over previously suggested algorithms through empirical evaluation. We started implementing our algorithm and the various baselines, but sadly, due to the short time, we did not complete the implementation in time for the rebuttal deadline. In particular, some baselines are extremely computationally demanding, and we would require additional time to perform statistically valid evaluations. We will update the final version of the paper with the results of this empirical evaluation.
>
> ### Questions
>
> 1/ This is indeed the case:  the optimal price depends on the context and varies with it. This is reflected in our expression of the regret as the optimal price is defined as the maximizer of the function $\pi(x_t, p)$, which depends on the context $x_t$.
>
> 2/ Concerning what is written in lines 277-278: both the papers directly mentioned in these lines share with us the assumptions of bounded context domain, parameter $\theta$, and noise, as those are the standard assumptions in the literature.
> - [1] make assumptions milder than ours, as they do not assume the c.d.f. $F$ to be Lipschitz. Instead, they rely on the half-Lipschitz nature of the reward function, which gives a worse rate $T^{3/4}$ overall. To understand why we need to assume Lispchitzness of $F$, we kindly refer to the answer 2 of the weaknesses in the response to reviewer Beqc.
> - [2] relies on estimating the parameter $\theta$ and $F$ in an explore-exploit fashion. They either achieve better regret rates under stronger assumptions, or worse rates under more similar assumptions. More precisely, their assumptions are stronger for the following reasons:
>
>     a/ To prove that the error from approximating the parameter $\theta$ is small they need to assume both the contexts to be i.i.d. and a lower bound on the eigenvalues of the covariance matrix of their variance. Our estimator for $\theta$ is built using a similar regression technique, but it does not require these assumptions (the contexts in our case can even be chosen adversarially), hence making the result more general.
>
>     b/ Furthermore, they assume the density of the noise to be lower-bounded by a constant $c$. The inverse of this quantity plays a role in the rate of the bound for the approximation of the CDF $F$, making it vacuous when $c$ is too small, and limiting the applicability of the result. Our estimate of the demand function relies on different tools, instead of the kernels estimators used in [2], hence does not require such an assumption.
>
>     c/ Another limiting assumption [2] introduce for their main algorithm is the strict monotonicity of the virtual value function (Assumption 2.1). This is necessary for them to ensure the uniqueness of the optimal price, which is needed to obtain small regret in their explore-exploit algorithm. Note how this is equivalent to asking the expected reward function $\pi$ to be unimodal, which, again, is a more restrictive setting than the one we consider. This last assumption is relaxed in Appendix F, however at the cost of sub-optimal regret rates of order $T^{3/4}$.
>
> In general, other rates can be obtained with stronger assumptions on the regularity of the distribution of the noise or the reward function, as summarized in Table 1. The algorithm we present is the one that achieves the best regret rates, especially optimal ones in the case of linear valuations, while at the same time requiring minimal assumptions on the noise distribution  (with the exception of the one in [1]) and allowing for adversarial contexts.
>
> ----
>
> [1] Xu, Jianyu, and Yu-Xiang Wang. "Towards agnostic feature-based dynamic pricing: Linear policies vs linear valuation with unknown noise." ICAIS, 2022.
> [2] Fan, Jianqing, Yongyi Guo, and Mengxin Yu. "Policy optimization using semiparametric models for dynamic pricing." JASA, 2024

---

> > ### Comment · Reviewer_EQLG · 2024-08-09
> > **Thank you for the clarifications**
> >
> > Thank you for the clarifications. I have no further questions. I increased my score to 6.

---

### Official Review · Reviewer_beqc · 2024-07-12

**Soundness:** 4
**Presentation:** 4
**Contribution:** 3
**Rating:** 8
**Confidence:** 4

**Summary:**

This paper studies the problem of online contextual pricing under a linear noisy valuation model, with the noise distribution **unknown** to the seller. This work proposes an "exploration-then-elimination" algorithm that achieves $O(T^{2/3})$ **optimal** regret under the assumptions of (1) stochastic feature sequence, (2) Lipschitz noise CDF, and (3) bounded feature norm and noise range. Besides, this work also studies the non-parametric $\beta$-Holder valuation model, and propose an algorithm that achieves $O(T^{d+2\beta/d+3\beta})$ regret.

**Strengths:**

1, The problem of contextual pricing problem has been extensively studied over the past few years. However, the minimax regret of linear noisy valuation model with unknown noise distribution is a long-existing open problem. Existing works leave a gap between $O(T^{3/4})$ and $\Omega(T^{2/3})$ under the assumption of Lipschitz noise CDF. This work closes the gap by showing that $O(T^{2/3})$ is the correct minimax regret, under certain assumptions.

2, This work also considers the non-parametric $\beta$-Holder valuation model, and propose an algorithm that achieves $O(T^{d+2\beta/d+3\beta})$ regret. This problem is relatively new to the community but it is important as well. In most of the cases, we cannot rely on the correctness of a linear model (although it is causally sound), and therefore this non-parametric model is of significance.

**Weaknesses:**

1, The algorithm, VAPE, proposed in this work is quite similar to the "Explore-then-UCB" algorithm in Luo et al. (2022, Neurips). Seems like the authors are unaware of this work at all. The only difference is that VAPE adopts a policy-elimination design in the second stage after uniformly pure exploration, while Luo et al. (2022) uses a UCB-style strategy. From the perspective of regret analysis, I do not think there are much difference between those two methods. I then went back to the previous work and looked into their regret analysis, and noticed that they were using a linear bandit model instead of a multi-armed bandit (which is also applicable), which brings an extra $\sqrt{\frac1{\Delta}}$ multiplier on the regret of the second stage, where $\Delta$ is the discretization grid size. In other words, by slightly modifying the "Explore-then-UCB" algorithm, it is able to achieve $O(T^{2/3})$ regret instead of $O(T^{3/4})$ as they claimed. Since both of your works share the same idea of uniformly-pure-exploration stage for an unbiased estimate of $x^\top\theta^*$, which actually originates from Fan et al. (2024, JASA), I am skeptical to the novelty of this work, especially from the perspective of methodology.

The authors are encouraged to discuss how their methods and analysis are different from Luo et al. (2022) in the rebuttal. This is the major concern that I (and other reviewers as I expect) would have.

2, The authors claim that it is required to assume Lipschitzness on the noise CDF, which is commonly applied except for Xu and Wang (2022, Aistats). It is worth noting that Xu and Wang (2022) have proposed an insight on the "Half-Lipschitz" nature of a pricing problem, indicating that the regret rate would not be different with or without Lipschitz assumption.

If this is not applicable, the authors are encouraged to explain in the rebuttal how the "Half-Lipschitz" nature does not work.



References:

Xu, Jianyu, and Yu-Xiang Wang. "Towards agnostic feature-based dynamic pricing: Linear policies vs linear valuation with unknown noise." International Conference on Artificial Intelligence and Statistics. PMLR, 2022.

Luo, Yiyun, Will Wei Sun, and Yufeng Liu. "Contextual dynamic pricing with unknown noise: Explore-then-ucb strategy and improved regrets." Advances in Neural Information Processing Systems 35 (2022): 37445-37457.

Fan, Jianqing, Yongyi Guo, and Mengxin Yu. "Policy optimization using semiparametric models for dynamic pricing." Journal of the American Statistical Association 119.545 (2024): 552-564.

**Questions:**

The authors are encouraged to respond to the following questions at a secondary priority (i.e. after kindly answering my questions in "Weaknesses").

1, Regarding Weakness 1, is there any necessity to apply a policy eliminiation strategy in stage 2 of VAPE? Can it be replaced by a (version of) UCB without impairing the regret rate?

2, On Page 5 Line 196, Please kindly explain the difference between your work and Fan et al. (2024) for valuation estimation methods in details.

**Limitations:**

The authors have discussed the limitations in their Conclusions. Overall, this is a theoretic work and there are not much societal impact involved. However, the authors are encouraged to discuss more on the societal aspect of the work, e.g. personalized pricing and unfairnesses.

---

> ### Author Rebuttal · Authors · 2024-08-06
>
> We thank the reviewer for the careful analysis.
>
> ### Weaknesses
> 1/ We thank the reviewer for providing us with this relevant reference.  As noted by the reviewer, both our work and that of [1] use the idea of sharing knowledge across contexts to improve the estimation of the c.d.f. $F$. However, our algorithm diverges significantly from theirs in three main aspects.
>
> First, our algorithm achieves the optimal regret rate, in contrast to [1]. [1] recast the problem as a perturbed linear bandit problem, which results in a **sub-optimal regret rate**, not merely as a proof artifact: the algorithm for perturbed linear bandits inherently exhibits regret that is linear in the dimension, here, the size of the discretization.
>
> Secondly, we believe that our algorithm is conceptually simpler. Instead of recasting the problem as a perturbed linear bandit in large dimensions, we phrase it as a simple, multi-arm bandit problem. Our expressions of the confidence intervals for the rewards are arguably more intuitive than the UCB in [1]. This simple formulation clarifies the sources of difficulty instead of obscuring them by reducing them to other intermediate problems, thereby helping the reader's understanding. We show that the problem is no more difficult than its non-contextual counterpart due to information sharing across contexts (highlighted in the proof of Claim 1), an idea absent from [1]: in contrast, the higher regret rate in [1] suggests that the contextual problem leads to higher regret. Additionally, our formulation helps generalizing our approach to other valuation models, as demonstrated with $\beta$-Hölder valuations.
>
> As a third point, we point out that [1] assumes i.i.d. contexts, with bounded eigenvalues of the covariance matrix. This assumption is removed in our work, and our algorithm can handle adversarial contexts.
>
> 2/ [2] rely on a black-box, expert aggregation algorithm, where each expert represents a possible value of the pair ($\theta$, $F$) on a discretized grid. They control the discretization error by noticing that posting a price slightly lower than the optimum price can only decrease the reward by the difference between the optimal price and this price. Thus, this half-Lipschitzness of the problem leads them to design an algorithm that essentially “rounds down” prices.
>
> By contrast, our algorithm first estimates the parameters, before running a bandit algorithm across the grid of prices that learns and shares information across contexts. This is only possible if the error made in the first phase in estimating $\theta$ implies a small error in the second phase in estimating $F(\delta)$ for values of $\delta$ over a grid. More precisely, if we knew exactly the value of  $x_t^{\top}\theta$, we could learn the value of $F(\delta)$ by posting a price $x_t^{\top}\theta + \delta$ without assuming Lipschitness of $F$. However, since we only know $x_t^{\top}\theta$ up to a small error $\epsilon$, we must assume that $F(\delta) \approx F(\delta \pm \epsilon)$ to learn $F(\delta)$.
>
> While our assumptions are stronger than that of [2], our regret rates are lower (and match the lower bound established for Lipschitz-continuous c.d.f.). Whether the rate of $T^{2/3}$ can be achieved without this assumption, or whether the minimax optimal rate differs when $F$ is not Lipschitz continuous, is an interesting question.
>
> ### Questions
> 1/ We agree that a UCB-type algorithm could be used instead of a Successive Elimination algorithm in the second phase of VAPE, provided that the expression for the Upper Confidence Bound for the rewards of the different prices be similar to that of Line 10 in Algorithm 2 or Algorithm 3. In this sense, there is no necessity to apply a successive elimination approach; however elimination-based methods are not conceptually more complex than a UCB-type of approach.
>
> 2/ We agree that our claim in Lines 185-186 is misleading. Indeed, some previous works have used the same method to estimate $g$, as noted by the reviewer. We sincerely apologize for conveying the wrong message.
>
> What we meant was that the remark that $g$ can be estimated using $1$-bit feedback allows us to decouple the estimation of $g$ and $F$ and estimate them in a straightforward manner, which seems to have been overlooked by previous works (that we were aware of). Closest to our method is probably [3]: the authors use the same method for estimating $g$, however, their estimation of $F$ strongly differs: they do it in an **explore-exploit fashion**, using **kernel estimators** that are fitted **on the samples already used to estimate $g$**. By contrast, we use piecewise-constant estimators, fitted in a regret-minimization sub-routine. We argue that our estimators are easier to interpret and that the assumptions necessary to establish our result are milder and less cumbersome than those of [3]. Moreover, explore-exploit approaches require i.i.d. contexts with strictly positive definite covariance matrices, while ours can handle adversarial contexts.
>
> However, as the reviewer pointed out, our approach for estimating the pair ($g$, $F$) is close to that of [1]. We therefore propose to remove this sentence altogether, and to add further detail on the distinction between our work, that of Luo et al (2022), and that of [3].
>
> **Limitations:** We acknowledge that the nature of the problem could raise fairness issues. We addressed some of these in the checklist under the “Broader Impacts” section. We are willing to move the paragraph to the main paper.
>
> ----
>
> [1] Luo, Yiyun, Will Wei Sun, and Yufeng Liu. "Contextual dynamic pricing with unknown noise: Explore-then-ucb strategy and improved regrets." NeurIPS, 2022
>
> [2] Xu, Jianyu, and Yu-Xiang Wang. "Towards agnostic feature-based dynamic pricing: Linear policies vs linear valuation with unknown noise." ICAIS, 2022.
>
> [3] Fan, Jianqing, Yongyi Guo, and Mengxin Yu. "Policy optimization using semiparametric models for dynamic pricing." JASA, 2024

---

> > ### Comment · Reviewer_beqc · 2024-08-12
> > **Thanks for your response**
> >
> > The reviewer thanks the authors for your insightful and exhaustive reply. For the comparison with [1], I somewhat agree that it is suboptimal to have a linear bandit algorithm, while yours are optimal. However, as I asked in Q1, by replacing your policy-elimination stage with a UCB, the algorithm is on the one hand optimal enough and on the other hand closer to their algorithmic design. However, since you are anyways the first to achieve optimal, and given that [1] had been accepted for Neurips 2022, I would like to support the acceptance of your paper.
> >
> > For the difference of assumptions comparing to [2], I am still skeptical if their half-lipschitzness is not applicable. In other words, I am not sure whether you really need $F(\delta) \approx F(\delta \pm \epsilon)$ which is stronger than it appears for a $T^{2/3}$ regret bound. For the estimates of $F$ and its difference with [3], I tend to believe that there are no substantial difference between (poly) kernels and discretizations. But I am happy to leave them for discussions beyond this paper.

---

### Decision · Program_Chairs · 2024-09-25

**Decision:**

Accept (poster)

**Comment:**

Executive Summary

The paper studies online contextual pricing. The model is that at each time step a context x_t \in R^d arrives. The algorithm sets a price o_t. The buyer's valuation is y_t = g(x_t) + \eta_t and the buyer buys if y_t >= p_t. The paper generally assumes that the noise is Lipschitz, and that the feature norm and noise range are bounded. It then presents results for two settings. The first setting is when g is linear, for this setting the paper gives a O(T^2/3) regret bound (improving the state of the art O(T^3/4) and matching a lower bound of Omega(T^2/3) shown in prior work ([30], AISTATS 2022).  The second setting is a non-parametric onw, which assumes that g is \beta-Holder, for which the authors show a O(T^{d+2\beta/d+3\beta) regret bound.  This non-parametric model is less-studied but of significance, as it provides some additional robustification. As far as I can tell the paper does not provide evidence of the tightness/non-tightness of the regret guarantees in the \beta-Holder valuation model.

Recommendation

I think this paper makes for a very nice NeurIPS paper, and closes the gap between upper and lower bound of the regret for a natural model of contextual pricing. I would therefore like to recommend it for acceptance (as a poster).
This recommendation is in line with the reviewers who were unanimously positive about the paper (8,7,6,6 ). I would ask the authors to keep the comments and suggestion from the reviewers in mind when preparing the final version.